# Thermogenetic neurostimulation with single-cell resolution

Yulia G. Ermakova[1,2], Aleksandr A. Lanin[3,4,5,6], Ilya V. Fedotov[3,5,6,7], Matvey Roshchin[8], Ilya V. Kelmanson[1], Dmitry Kulik[3,†], Yulia A. Bogdanova[1], Arina G. Shokhina[1], Dmitry S. Bilan[1,2], Dmitry B. Staroverov[1], Pavel M. Balaban[8], Andrei B. Fedotov[3,4,5,6], Dmitry A. Sidorov-Biryukov[3,5,6], Evgeny S. Nikitin[8], Aleksei M. Zheltikov[3,4,5,6,7] & Vsevolod V. Belousov[1,2,9]

Thermogenetics is a promising innovative neurostimulation technique, which enables robust activation of neurons using thermosensitive transient receptor potential (TRP) cation channels. Broader application of this approach in neuroscience is, however, hindered by a limited variety of suitable ion channels, and by low spatial and temporal resolution of neuronal activation when TRP channels are activated by ambient temperature variations or chemical agonists. Here, we demonstrate rapid, robust and reproducible repeated activation of snake TRPA1 channels heterologously expressed in non-neuronal cells, mouse neurons and zebrafish neurons in vivo by infrared (IR) laser radiation. A fibre-optic probe that integrates a nitrogen−vacancy (NV) diamond quantum sensor with optical and microwave waveguide delivery enables thermometry with single-cell resolution, allowing neurons to be activated by exceptionally mild heating, thus preventing the damaging effects of excessive heat. The neuronal responses to the activation by IR laser radiation are fully characterized using $Ca^{2+}$ imaging and electrophysiology, providing, for the first time, a complete framework for a thermogenetic manipulation of individual neurons using IR light.

[1] Shemyakin-Ovchinnikov Institute of Bioorganic Chemistry, Russian Academy of Sciences, Moscow 117997, Russia. [2] Pirogov Russian National Research Medical University, Moscow 117997, Russia. [3] Physics Department, International Laser Center, M.V. Lomonosov Moscow State University, Moscow 119992, Russia. [4] Department of Physics and Astronomy, Texas A&M University, College Station, Texas 77843, USA. [5] Russian Quantum Center, ul. Novaya 100, Skolkovo, Moscow Region 143025, Russia. [6] Kazan Quantum Center, A.N. Tupolev Kazan National Research Technical University, 420126 Kazan, Russia. [7] Kurchatov Institute National Research Center, Moscow 123182, Russia. [8] Institute of Higher Nervous Activity and Neurophysiology, Moscow 117485, Russia. [9] Institute for Cardiovascular Physiology, Georg August University Göttingen, D-37075 Göttingen, Germany. † Present address: Zaporizhya State Engineering Academy, 69006 Zaporizhzhya, Ukraine. Correspondence and requests for materials should be addressed to A.M.Z. (email: zheltikov@physics.msu.ru) or to V.V.B. (email: vsevolod.belousov@gmail.com).

Optogenetic methods allow specific neuronal subsets to be operated by light with high precision in space and time, and are important tools helping to understand the mechanisms of functioning of the brain and peripheral neural system[1,2]. Genetic targeting enables spatial resolution by restricting expression of light-activated ion channels to genetically defined subpopulations of neurons. Programmable light patterns can further improve the spatial resolution and add temporal control to the optogenetic toolbox[3]. Optogenetic tools allow both activation and silencing of neurons by means of activation of depolarizing cationic channels or hyperpolarizing anionic pumps, respectively[1,2,4]. The broad range of model organisms suitable for optogenetics and continuously upgraded light-gated channels make this toolbox more and more popular and affordable[5].

While enabling revolutionary new experiments for dissecting neuronal circuits and modulating animal behaviour, optogenetics is not free from serious limitations and difficulties. First, the ion channels commonly used in optogenetic studies need visible light for activation. Extensive ongoing research aimed at designing red-shifted channelrhodopsins and halorhodopsins is still far from identifying effective ways towards deeper penetration depths for the activation radiation, as visible light is still being used in most of the studies to activate the most popular ChR2 (refs. 6–8). As a consequence, stimulation of neurons in optically non-transparent animals, such as mice, is often invasive, involving implantation of optical fibres. As a serious side effect, the activating light acts on intrinsic visual receptors of the animals[9], which may cause partial blindness and unwanted brain circuit activation. Moreover, despite recent progress in making channelrhodopsins with high single-channel conductance[10], the low conductance of most ChRs (40–60 fS for ChR2)[11] necessitates high channel expression levels and high activation light intensities, which are very likely to induce rarely considered harmful phototoxic effects.

Thermogenetics offers a promising alternative approach, based on heat-activated cation channels from the transient receptor potential (TRP) family[12]. The TRP channels are three orders of magnitude more conductive than ChR (ref. 13) and can be activated by heat induced with infrared (IR) light[14] or magnetic particles in a strong magnetic field[15]. Thermogenetic approaches have been already successfully demonstrated in studies on ectotherms, such as *Drosophila* and zebrafish, as well as endotherms, such as mice[16]. With all this success, thermogenetics, as a new technology for neuroscience, is still in its infancy. Indeed, most of the thermogenetic experiments with *Drosophila* neural circuits used air temperature changes to activate the TRP channels[17–20]. In recent experiments with zebrafish *Danio rerio*, chemical agonists of TRP channels were used for *in vivo* neuron activation[21]. However, the agonist washout needed for this approach severely limits the speed of sequential activation. As a helpful alternative, IR laser radiation has been shown to enable activation of neurons in flies with relatively high spatial and temporal precision[14,22]. For demonstration purposes, these experiments were performed on *Drosophila* native fly thermoreceptors, dTRPA1. However, the TRP channels are complex and subject to many post-translation modifications and protein–protein interactions, which can modify channel activity[23]. As many of the post-translation modification sites and interactions are species-specific, the use of recombinant channels from evolutionary distant species would significantly lower the risk of unwanted channel modulation and increase the resolution of activation patterns. The improvement of the thermogenetic toolbox should therefore start with a systematic search and characterization of suitable IR-sensitive channels that would enable robust activation of neurons with minimal risk of unwanted interactions with the cellular and tissue environment. An equally important task is to develop efficient photonic activation approaches that would overcome the limitations of chemical agonists or ambient temperature changes as thermogenetic stimuli. This task is not limited to identifying most suitable heating sources, but includes the development of appropriate tools for a high-precision local thermometry that would help to finely adjust the incoming heat flow.

In the present study, we use TRPA1 channels from snakes as a thermogenetic stimulator. Many snake species use a special heat-sensitive organ, called a 'pit', to distantly localize a prey. The pit organ is innervated with receptor neurons expressing the TRPA1 channels, which serve as molecular heat sensors[24]. In this work, the snake TRPA1 channels are used for thermogenetic activation of mouse neurons and induction of zebrafish larvae behavioural response. Neurons expressing rattlesnake TRPA1 were activated in our experiments using IR-laser-controlled heating with a fibre-optic probe integrated with a quantum sensor used for *in situ* high-resolution thermometry, thus enabling an exceptionally mild, precisely controlled heating of the tissue. With this combination of methods, we carried out a systematic, in-depth characterization of the performance of TRPA1 channels in neurons, including accurate measurements of the pertinent $Ca^{2+}$ dynamics and electrophysiological analysis of the responses. Our study demonstrates that the snake TRPA channels are ideally suited as a tool for thermogenetics, providing a carefully validated kit of molecular and optical instruments as a route toward mature thermogenetic technologies for neuroscience.

## Results

**Visualization of TRPA1 activation.** Visualization of optogenetic actuators inside living cells allows estimation of their expression level, as well as their distribution among and within cells. The most obvious way towards this goal is to fuse an ion channel with a fluorescent protein (FP). This strategy was successfully implemented for the majority of light-gated channels, such as ChR variants[5,25], helping to monitor both localization and the expression level of the channel. However, linking an FP to either amino- or carboxy-termini could negatively affect channel function. Another option is to leave the channel intact, but to add an FP as an expression marker using either an IRES sequence[26,27] or a co-transcriptional self-cleavage P2A linker[28]. This strategy ensures that the channel is intact and operates in the native mode. Moreover, a P2A sequence allows the expression level to be estimated. However, the location of the channel cannot be determined without specific antibodies. Finally, inserting an epitope, such as a His-tag, into the surface loop of the protein makes it possible to visualize the channel through immunostaining[15,16].

To find an optimal labelling strategy for the snake TRPA1 channels, we tested the above-mentioned procedures (Supplementary Note 1, Fig. 1a, Supplementary Figs 1–5).We generated two expression vectors: one encoding EGFP (refs 29,30) downstream to the *Crotalus atrox* TRPA1 (caTRPA1, ref. 24) under the control of an IRES element (caTRPA1-IRES-EGFP) and the other encoding caTRPA1 fused in frame to red FP tdTomato[31] via a linker containing a P2A cleavage site (caTRPA1-P2A-tdTomato) (Supplementary Fig. 1). A feature of the IRES element is that the expression of the second open reading frame (ORF) is about tenfold less intense than that of the first ORF upstream of the IRES. However, the ion channel amino-acid sequence remains completely intact. The caTRPA1-P2A-tdTomato ensures the most reliable information about the expression levels of both caTRPA1 and FP because they both are synthesized as a single polypeptide chain with subsequent cleavage.

Multiple fusion constructs of caTRPA1 with FPs gave us important information about the subcellular localization of the channels (Fig. 1a, Supplementary Note 1, Supplementary Figs 2–4) that appeared to be predominantly plasma membrane. However, the non-fused channels demonstrated much better functional performance (Supplementary Note 1, Supplementary Fig. 5). Therefore we decided for using intact channels in further experiments.

Similar results were obtained for another snake channel, *Elaphe obsoleta lindheimeri* TRPA1 (eolTRPA1) (ref. 24) (Supplementary Fig. 2c, Supplementary Fig. 4c).

**Expression of caTRPA1 in mammalian cells.** Mammalian cultured cell lines and primary neurons could be the most suitable systems to study the key parameters of snake TRPA1 channel activity. However, the activation threshold of the known snake TRPA1 channels is either lower than $T_{ON} \approx 27.6 \pm 0.9\,°C$ (for caTRPA1) (ref. 24) or equal to $T_{ON} \approx 37.2 \pm 0.7\,°C$

(for eolTRPA1) (ref. 24), the temperature of the cells maintained in a $CO_2$ incubator. Therefore, we expected some toxicity associated with $Ca^{2+}$ entry via the TRPA1 at constantly elevated temperatures. We compared the heat- and agonist-induced $Ca^{2+}$ responses in caTRPA1-expressing HEK293 cells cultured at different temperatures, viz., 25, 30 and 37 °C. Cell growth at 25 °C led to a four- to tenfold decrease in the transfection efficiency and lower expression levels of TRPA1, R-GECO1.1 (ref. 32), and GCaMP6s (ref. 33). This decreased the stimulation efficiency (Fig. 1b,c) and the amplitude of the $Ca^{2+}$ signal.

Surprisingly, cell cultivation at 30 or 37 °C does not lead to permanent activation of the caTRPA1. After transfer of the cells expressing caTRPA1 from high temperature to the microscope stage at the room temperature (RT), temperature conditioning at 25 °C within an hour is needed to make the cells respond to either heating to 30 °C or TRPA1 agonist allyl isothiocyanate (AITC) (Fig. 1b,c). Similar effects were observed for eolTRPA1

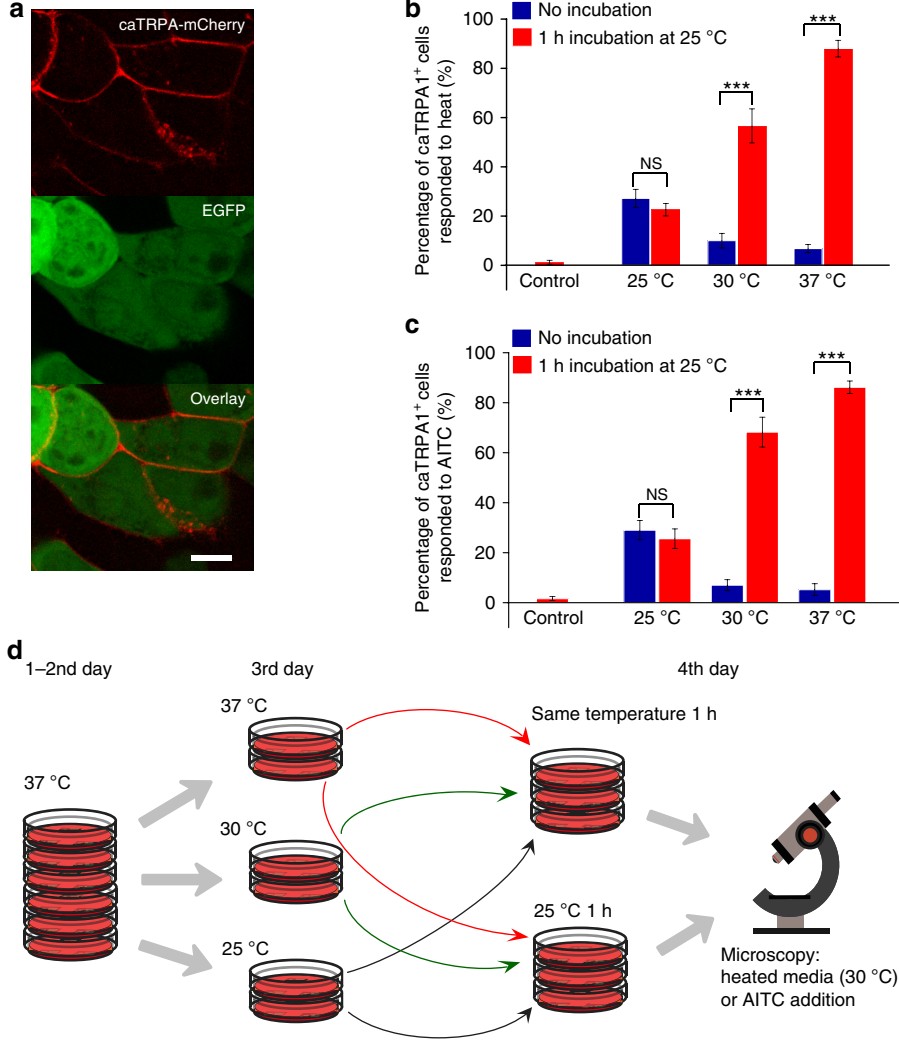

**Figure 1 | HEK293 cells expressing caTRPA1 grown at different temperatures respond to a rapid rise of the temperature or agonist AITC.** (**a**) Localization of caTRPA1-mCherry. Scale bar, 10 μm. (**b**) HEK293 cells were co-transfected with caTRPA1-IRES2-EGFP and $Ca^{2+}$ sensor R-GECO1.1. After 36–48 h, cells were transferred to another incubator for temperature conditioning at 25, 30 or 37 °C for 18–26 h. Next, half of the cells were kept for additional 1 h at the same temperature and another half of the cells were kept at 25 °C. Then all cells were transferred to the stage of the microscope preheated to 25 °C and activated by the addition of warm medium under control of electrode thermometer. Final temperature was 30 ± 1 °C. Control cells express only R-GECO1.1 indicator. (**c**) Response of cells to 200 μM AITC. For each experimental condition in (**b**) and (**c**), 50 or more cells were analysed in six experiments. ns—not significant. ***$P < 0.001$, paired *t*-test (two-tailed) for two-group comparisons. The error bars represent the s.e.m. (**d**) Scheme of the experimental workflow.

conditioned below the threshold at 30 °C (Supplementary Fig. 6) and heated to 38 °C. This indicates that either the channels have a mechanism for self-protection against remaining in a constantly open state or the final channel folding stage requires subthreshold temperatures. This feature makes mammalian cell systems suitable for studying snake TRPA1 activation properties.

**Determination of caTRPA1 temperature threshold.** Thermogenetic channels are activated with heat. IR laser radiation offers a robust and flexible method to induce heat in a well-controlled, precisely localized fashion, with switching between ON and OFF states of a thermosensitive channel performed at much higher speed than with any of the available techniques used for ambient temperature variations. Laser-induced heating can be accurately controlled by finely adjusting laser radiation intensity. This advantage of laser heating is of crucial importance for *in vivo* studies as living organisms widely differ in ranges of tolerable temperature variations, graded from the normal through elevated to the toxic level. Moreover, the method of heat delivery may affect the channel threshold temperature[17,24,34].

The wavelength of laser radiation $\lambda$ is one of the the key parameters controlling laser-induced heating of a cell and the surrounding tissue. To provide efficient absorption of laser radiation by surrounding tissues and to simultaneously minimize phototoxic effects that may be caused by direct absorption of laser light by molecules within the targeted living cells, we chose to work at a wavelength of 1,440 nm, which hits the local maximum of water absorption within the 200–1,900-nm range[35]. The absorption coefficient at this wavelength estimated as $\mu_a \approx 32$ cm$^{-1}$. Importantly, this wavelength falls within the transparency window of glass and quartz components of most optical systems, allowing the light coming from a laser source to be manipulated in a flexible and reliable fashion.

Our laser-based system combining IR excitation with fluorescent microscopy and high-resolution quantum thermometry of activated cells is sketched in Fig. 2a. In this experimental scheme, the frequency-tunable output of the Ti:sapphire-laser-pumped femtosecond optical parametric oscillator (OPO) is loosely focused with a long-focal-length lens into a spot 670 μm in diameter, providing a uniform irradiation of a large number of TRPA-expressing cells within the sample. The transverse profile of the laser beam measured within the sample plane is shown in Fig. 2b. For highest efficiency of laser heating, the central wavelength of the OPO output is tuned to the local maximum of water absorption at around 1,440 nm. Ultrashort ($\approx 100$ fs) IR pulses were used for laser irradiation[36,37]. Continuous-wave laser sources delivering 5-mW radiation at 473 and 532 nm were employed for multicolour fluorescent imaging using R-GECO1.1, EGFP, GCaMP6s and tdTomato as fluorescent biomarkers.

A fibre-optic probe with a 300-μm-diameter nitrogen− vacancy (NV) diamond crystal attached to the tip of the fibre (Fig. 2a) adds a unique modality to our experiments, enabling accurate *in situ* temperature measurements with single-cell resolution[38–40] by means of ODMR (see the Methods section for a detailed description of fibre-based thermometry implemented as a part of our experimental approach). In Fig. 2d, we present the difference $\Delta T = T - T_0$ between the temperatures $T$ and $T_0$ measured with and without laser irradiation of the sample as a function of the laser wavelength $\lambda$. The $\Delta T(\lambda)$ dependence (the solid line in Fig. 2d) strongly correlates with the spectrum of water absorption (the dashed line in Fig. 2d), showing that absorption of IR light by water dominates laser-induced heating in our experiments, with the heating rate per unit laser power within the studied range of laser intensities estimated as $0.25 \pm 0.01$ K mW$^{-1}$ for $\lambda \approx 1,440$ nm

(Fig. 2e). The transverse profile of the laser-induced heating $\Delta T$ resolved by scanning the fibre-optic sensor across the laser beam confirms that the radial distribution of $\Delta T$ within the laser beam corresponds to the radial field intensity profile, with the maximum of laser excitation strongly confined to the centre of the laser-irradiated area.

The laser imaging system with *in situ* fibre-optic thermometry provides all the necessary functionality to analyse the response of TRPA1-expressing cells to 1,440-nm laser radiation and to define the activation thresholds of caTRPA1 and eolTRPA1. HEK293 cells co-expressing caTRPA1-IRES-EGFP and R-GECO1.1 were kept overnight at 26 °C and subjected to irradiation with femtosecond IR laser pulses, delivered by the OPO operating in the quasi-continuous-wave (quasi-cw) mode with the 78-MHz pulse repetition rate. Rapid temperature rise from subthreshold temperatures to 30.5 °C led to channel opening and build-up of elevated concentrations of Ca$^{2+}$ in the cytoplasm. When the laser beam was blocked, R-GECO1.1 fluorescence gradually faded away (Fig. 3a), indicating channel closure and Ca$^{2+}$ pumping out from the cytoplasm.

Next, we defined the temperature thresholds for caTRPA1 and eolTRPA1 by adjusting the IR laser irradiation in such a way as to increase the temperature in a stepwise manner, as shown in Fig. 3b,c. The activation thresholds were found to be $T_{ON} \approx 27.8 \pm 0.6$ °C and 38.5 ± 0.7 °C for caTRPA1 and eolTRPA1, respectively. Control cells expressing only Ca$^{2+}$ indicator did not respond to the IR irradiation (Fig. 3d), showing that ultrashort IR laser pulses can efficiently activate snake TRPA1 channels without exerting any detectable influence on normal, non-TRPA1-expressing cells in thermogenetics experiments.

Alternatively, we determined the temperature thresholds as the point of intersection between linear fits to baseline and the steepest component of the Arrhenius profile[24] (Supplementary Fig. 7a,b). The values obtained were 27.4 ± 0.4 and 37.9 ± 0.8 °C for caTRPA1 and eolTRPA1, respectively.

Repeated heating elevated the level of Ca$^{2+}$ again, with the entire cycle fully reproduced each time the laser beam was turned on and then off again (Fig. 3e,f, Supplementary Fig. 7c), indicating full functionality of caTRPA1 in the plasma membrane of mammalian cells.

Our experiments also showed that Ca$^{2+}$ dynamics in the cytoplasm can be efficiently tailored towards the desired shape by an appropriate modulation of the laser-driven heat release. Faster heating was observed to induce faster and more significant build-up of Ca$^{2+}$ elevation, while heating just above activation thresholds led to slower build-up of Ca$^{2+}$. With various modulation patterns applied to the intensity of ultrashort laser pulses as a function of time, we were able to tailor the traces of the Ca$^{2+}$ response from the cells to various waveforms (Supplementary Fig. 7d). As a limiting factor, slow (0.5–1 °C min$^{-1}$) heating of the cells from subthreshold to above-threshold temperatures did not lead to TRPA1 activation. The nature of such behaviour of channels is still to be understood.

**Activation of mammalian neurons using the snake TRPA1.** Several studies demonstrate the possibility of temperature-operated neuronal and muscle activity[41–47]. However, whether non-mammalian TRPA channels can be used for mammalian neuron activation remained unclear. Moreover, whereas numerous reports describe electrophysiological properties of neurons during optogenetic activation, all studies using thermogenetic activation demonstrate mostly *in vivo* behavioural outcomes without any measure of the neuronal response properties. Most of the previous studies with rare exceptions[14] did not use the IR photonics for stimulation, but rather relied on ambient temperature changes or

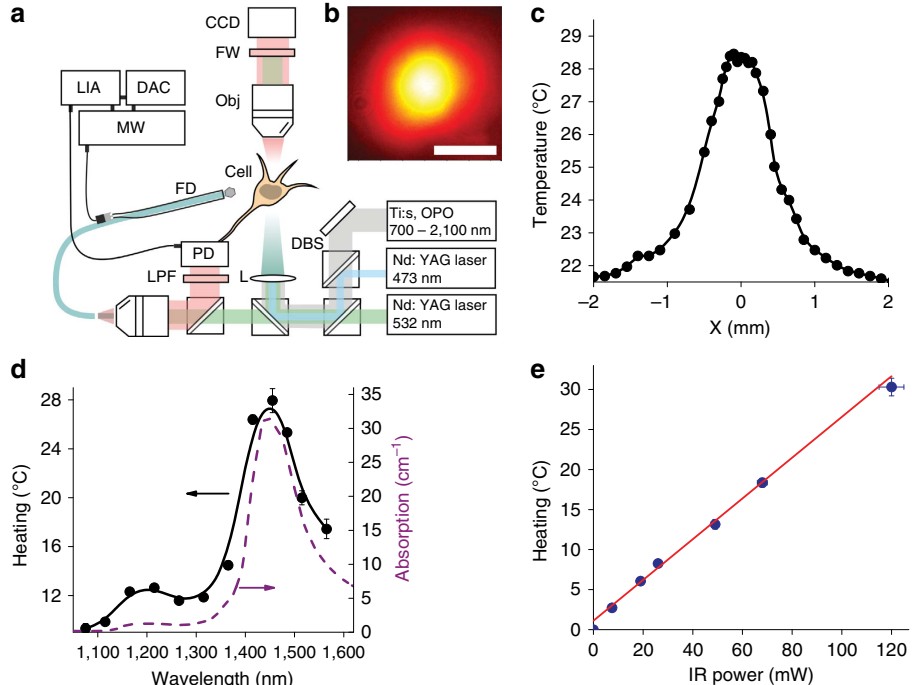

**Figure 2 | Local heating and quantum thermometry of living cells.** (**a**) Laser thermogenetics and fluorescent imaging of single cells: Ti:s, mode-locked Ti:sapphire laser; OPO, femtosecond optical parametric oscillator; Obj, microscope objective; FW, filter wheels; DBS, dichroic beam-splitting cube; LPF, long-pass filter; FD, fibre probe with a NV-diamond quantum temperature sensor; MW, microwave source; LIA, lock-in amplifier; DAC, data acquisition circuit; CCD, CCD camera. (**b**) Transverse profile of the IR laser beam. The scale bar, 500 μm. (**c**) Spatial profile of temperature distribution upon 36 mW 1,440 nm laser irradiation measured using diamond positioned on the tip of the optical fibre (details in the Methods section). Basal temperature is 19.5 °C. (**d**) Absorption spectrum of water (dashed line) versus the temperature change $\Delta T = T - T_0$ induced in the sample by 104-mW laser radiation measured as a function of the laser wavelength $\lambda$ (solid line), where $T$ and $T_0$ are the temperatures of the sample with and without laser radiation. (**e**) The temperature change $\Delta T$ measured as a function of the IR laser power with $\lambda = 1,440$ nm.

chemical agonists. To close these gaps in the existing knowledge, we analysed $Ca^{2+}$ dynamics and electrophysiological properties of the thermogenetic neuronal response.

Expression of TRPA1-IRES-EGFP in neurons resulted in a very weak and cell-variable EGFP fluorescence signal. In contrast, TRPA1-P2A-tdTomato demonstrated strong fluorescence in neurons. We therefore used co-transfection of TRPA1-P2A-tdTomato with GCaMP6s to visualize $Ca^{2+}$ dynamics in neurons upon TRPA1 activation (Fig. 4).

In our experiments with IR laser radiation, the diameter of the laser beam ($\approx 60\,\mu m$) was comparable to the size of a single neuron, thus enabling activation of individual neurons without affecting the neighbouring cells (Fig. 4a,e,f). Further heating leads to activation of the neighbouring neuron (Fig. 4e,f). Similar to agonist stimulation, the IR stimulation of caTRPA1-expressing neurons induced high-amplitude $Ca^{2+}$ transients (Fig. 4b,c). The amplitude of the $Ca^{2+}$ signal from eolTRPA1 was lower than that of caTRPA1 in both IR stimulation and AITC stimulation. Control cells transfected with tdTomato and GCaMP6s responded to neither IR nor AITC (Fig. 4b).

A persistent $Ca^{2+}$ signal observed within the entire period of photostimulation (Fig. 4c) indicates that a certain fraction of thermosensitive channels remain open all this time, allowing the dynamics of $Ca^{2+}$ in the cytoplasm to be efficiently controlled with proper timing of thermogenetic stimuli. While the extensive literature[41,43–47] suggests that, with sufficiently high laser intensities, laser activation of neurons is possible without heterologous heat-sensitive channels, our experiments with control neurons that do not express heterologous TRPA1 clearly show that, when the laser power is kept at a moderate level (below 90 mW for the 1,440-nm quasi-cw output of the

femtosecond OPO in our experiments), giving rise to well-controlled (and accurately monitored with fibre-optic sensors) heating within the range of temperatures from 25 to 40 °C, the IR-laser stimulation-evoked $Ca^{2+}$ build-up can be completely avoided (Supplementary Fig. 8).

It should also be noted that the TRPA1 temperature activation threshold in neurons may differ from the activation threshold of these channels in non-excitable HEK293 cells. Our direct measurements show (Fig. 4c,e,f) that the activation threshold of caTRPA1 in neurons is $T_{ON} \approx 28.5 \pm 1$ °C. Since the basal temperature of zebrafish maintenance in fish facilities is generally maintained between 26 and 28.5 °C (refs 48–50), these measurements prove that TRPA1 channels offer a powerful tool to study *D. rerio* neurophysiology.

**Activation of neurons using pulsed IR laser radiation.** When operated in the pulsed mode, the IR laser sources of ultrashort pulses provide additional benefits for carefully controlled neuron photoactivation, enabling a fine adjustment of irradiation powers, doses, times and pulse repetition rates, as well as helping to reduce the phototoxic effects of laser radiation. In our experiments, pulsed irradiation of samples was performed by using a chopped OPO output, delivering trains of femtosecond IR pulses with a tunable pulse-train duration $\tau_t$ and pulse-train repetition rate $f_t$. When applied to the caTRPA1-expressing neurons, such IR pulse trains were found to induce $Ca^{2+}$ transients at their repetition rate $f_t$ (Fig. 4g,h). No fluorescent response was observed from the control neurons.

Fast response is one of the key criteria defining whether or not an ion channel can be used for opto- or thermogenetic manipulation of neurons. Tunable trains of femtosecond pulses

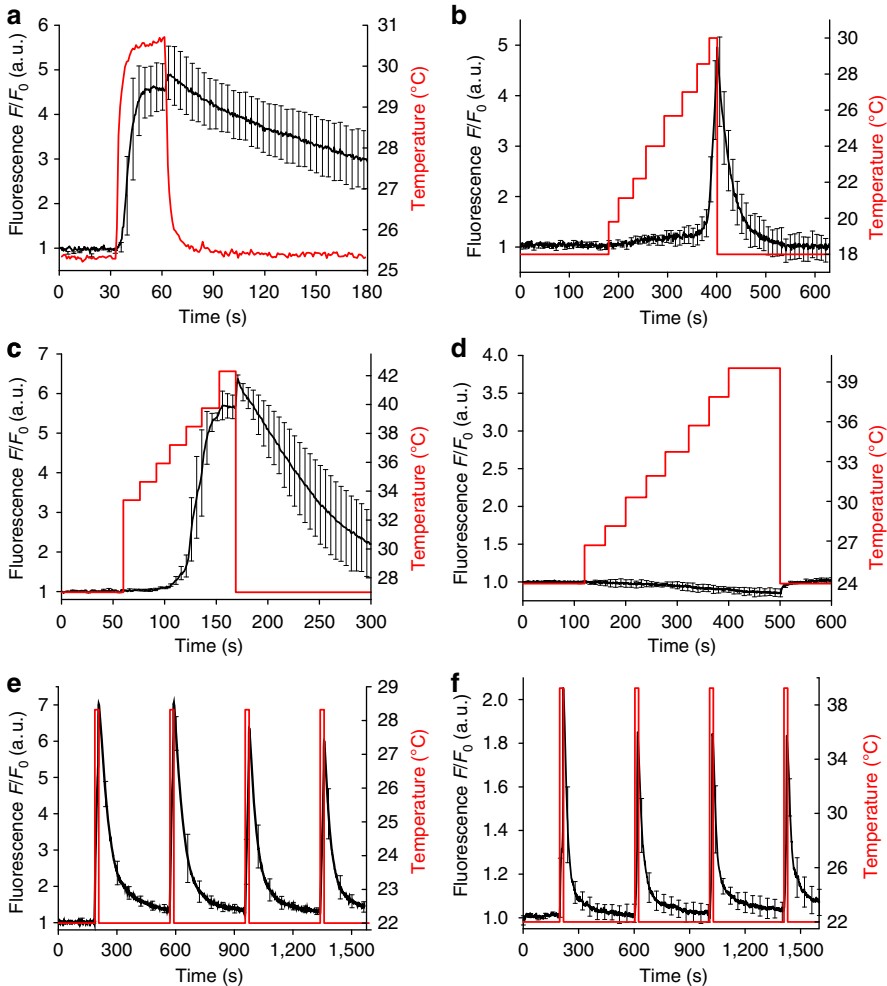

**Figure 3 | Activation of snake TRPA1 in cells expressing TRPA1-IRES-EGFP using femtosecond IR laser pulses. (a)** R-GECO1.1 fluorescence (black line) reflects $Ca^{2+}$ dynamics in the cytoplasm with the 20 mW laser beam turned on at $t \approx 30$ s and off at $t \approx 60$ s. **(b,c)** With the temperature of HEK293 cells expressing snake TRPA1 increased in a stepwise fashion using properly adjusted IR laser radiation, the activation thresholds of caTRPA1 **(b)** and eolTRPA1 **(c)** were determined. **(d)** A similar heating of control cells does not induce $Ca^{2+}$ elevation. **(e)** Repeated caTRPA1 activation cycles using 27 mW laser. **(f)** Repeated eolTRPA1 activation cycles using 68 mW laser. The fluorescence signal is averaged over 11 **(a)**, 5 **(b–d)**, 30 **(e)** and 17 **(f)** cells. The black line is the fluorescence response. The red line is the temperature in the medium. The error bars represent the s.e.m.

provided by wavelength-tunable OPOs are ideally suited for time-resolved studies of caTRPA1 opening in neurons. In our experiments, isolated 10-ms trains of 1,440-nm femtosecond laser pulses were used to activate the caTRPA1-expressing neurons. The build-up and decay of $Ca^{2+}$ elevation in the cytoplasm was then studied (Fig. 4i) by means of fluorescence imaging. The typical activation time for this system was estimated as $\sim 10$ ms. Fluorescence decay was found to be exponential, $F/F_0 \sim \exp(-t/\tau_d)$, with $\tau_d \approx 3.5 \pm 0.1$ s (green line in Fig. 4i). This exponential decay is most likely to be related to a dissociation of $Ca^{2+}$ in GCaMP6s, rather than the dynamics of TRPA1 closing.

The caTRPA1 channels thus open on the millisecond timescale, which is comparable with typical response times of light-sensitive channels used in optogenetics, such as channelrhodopsins. Although these experiments strongly suggest that $Ca^{2+}$ biosensors could be very useful as instruments for visualizing the activity of neurons in response to thermogenetic stimulation, association and dissociation of $Ca^{2+}$ in the probe could be a potential source of errors in $Ca^{2+}$-imaging-based assessment of channel dynamics. This issue was addressed through direct studies of channel dynamics by means of electrophysiological analysis of the snake TRPA1-expressing neurons.

**Electrophysiology and imaging of cultured mammalian neurons.** To test the ability of the caTRPA1 and eolTRPA1 channels to evoke action potentials, we recorded the response of cultured hippocampal/neocortical neurons transfected with caTRPA1-P2A-tdTomato to isolated millisecond pulses of IR radiation applied to the soma of neurons, as well as to the proximal parts of axons and dendrites (Fig. 5, Supplementary Fig. 9). Laser pulses with a central wavelength of 1,050 or 1,342 nm were produced by chopping the cw output of an ytterbium fibre laser or by a diode-pumped solid-state laser. IR radiation was delivered to the targeted areas within the neurons under study through an optical fibre with a core diameter of 50 μm and a numerical aperture of 0.22.

Laser pulses of 10 ms duration and energy $E$ at a wavelength of 1,342 nm were found to evoke single action potentials (APs) in both caTRPA1-expressing neurons ($E \approx 1.0$ mJ, $n = 5$, Fig. 5a, top trace) and eolTRPA1-expressing ($E \approx 1.7$ mJ, $n = 5$; Fig. 5b, top trace) neurons kept at basal temperatures of 27 °C and 35.5 °C, respectively. Similar responses have been obtained using 30-ms pulses of the ytterbium fibre laser at 1,050 nm that induced at least one action potential in each of the recorded caTRPA1 + ($E \approx 27$ mJ, $n = 5$; Supplementary Fig. 10b, middle trace) and

eolTRPA1 + cells ($E \approx 45$ mJ, $n = 5$; Supplementary Fig. 10f, middle trace), evoking no action potential in the control caTRPA − and eolTRPA1 − cells ($n = 5$) (Supplementary Fig. 10i,j). Longer, 50 ms, pulses normally induced more than one action potentials (Supplementary Fig. 11).

Laser pulses of low energy were found to evoke subthreshold voltage responses in both caTRPA1-expressing ($E \approx 9$ mJ, $9.8 \pm 1.3$ mV, $n = 5$; Supplementary Fig. 10b, top trace) and eolTRPA1-expressing ($E \approx 27$ mJ, $12 \pm 3.4$ mV, $n = 5$;

Supplementary Fig. 10f, top trace) neurons kept at basal temperatures of 27 °C and 35.5 °C, respectively. The response induced by laser pulses with $E \approx 9$ mJ and 27 mJ in the control caTRPA1 − and eolTRPA1 − (tdTomato alone) neurons, respectively, was much weaker ($2.4 \pm 0.5$ mV for caTRPA1 − neurons, $n = 5$; Supplementary Fig. 10b, bottom trace and $2.5 \pm 0.7$ mV for eolTRPA1 − neurons, $n = 5$; Supplementary Fig. 10f, bottom trace) than the response of caTRPA1 + and eolTRPA1 + neurons to subthreshold stimulation by laser pulses with the same power.

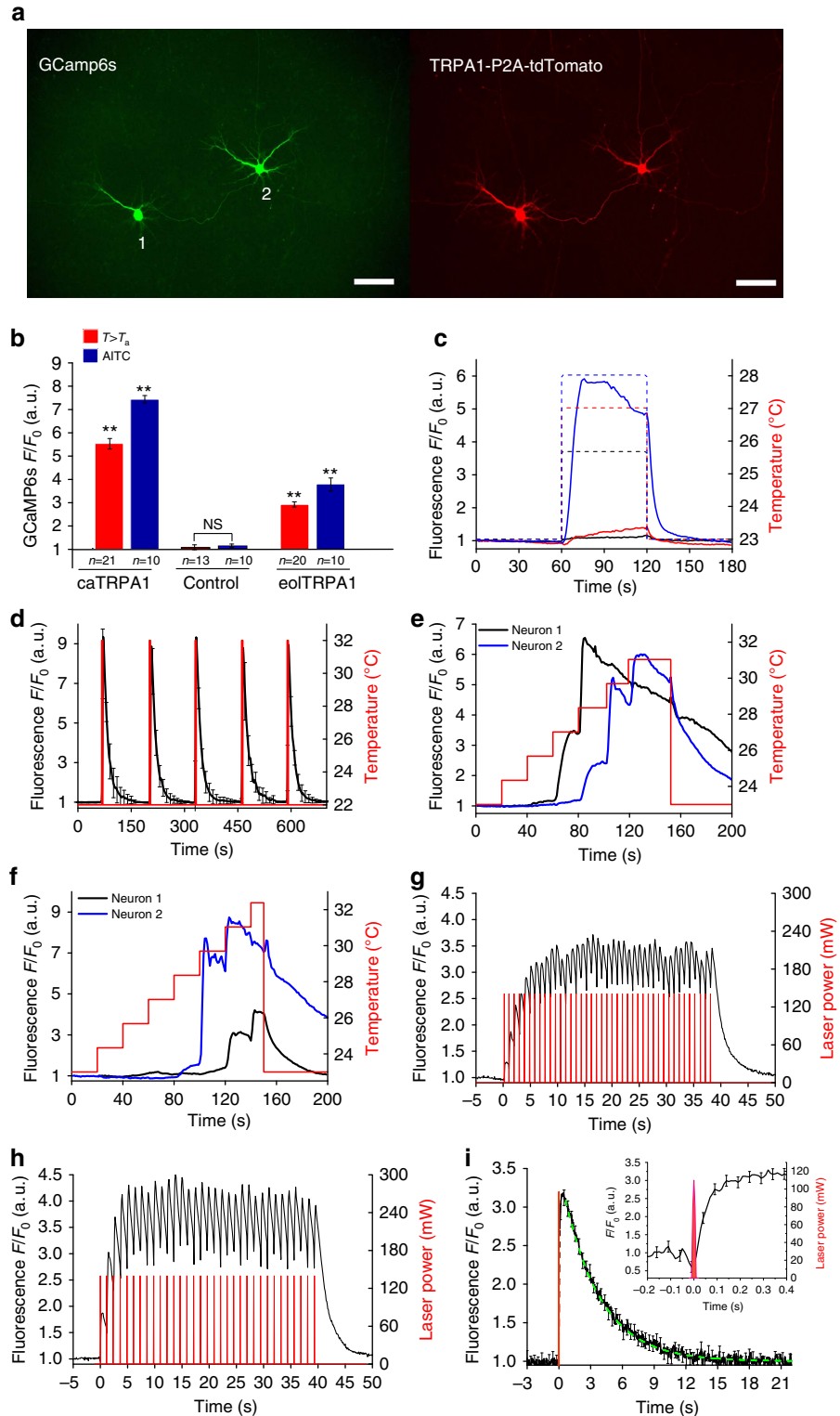

Pulsed IR stimulation of the neurons held in the voltage-clamp mode at a holding potential of $-90\,\text{mV}$ gave rise to a transient inward current, which rapidly faded when stimulation was over (Supplementary Fig. 10c). Calibration experiments demonstrated that, with the 30-ms 1,050-nm laser pulses of 9 and 27 mJ applied to the caTRPA1 channels and laser pulses of 27 and 45 mJ applied to the eolTRPA1 channels, the temperature induced by laser heating was above the activation threshold of caTRPA1 (Supplementary Fig. 9, Supplementary Fig. 10d) and eolTRPA1 channels (Supplementary Fig. 9, Supplementary Fig. 10g).

Remarkably, in experiments where a 25 or 50 Hz trains of 10-ms 1,342-nm pulses with $E \approx 1\,\text{mJ}$ were used to stimulate caTRPA1-expressing neurons, trains were found to induce a clear phase-locked response (Fig. 5c–e) in action potentials generation. The frequency of thermogenetically-evoked reliable phase-locked

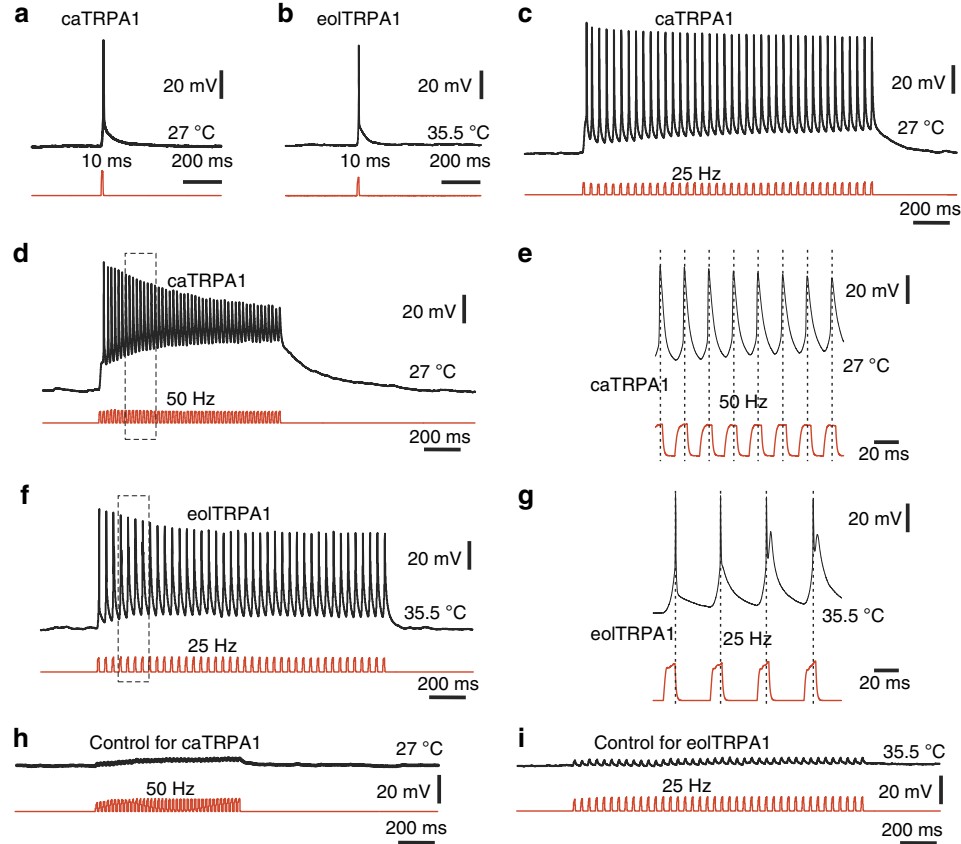

**Figure 5 | Thermogenetic induction of phase-locked responses in mammalian cultured neurons expressing TRPA1-P2A-tdTomato constructs.**
(**a**) Example of a single AP induced by 10-ms laser pulse with a central wavelength of 1,342 nm (energy $\sim 1.0$ mJ) in a caTRPA1+ neuron resting at 27 °C. (**b**) Example of a single AP induced by 10 ms IR stimulation (energy $\sim 1.7$ mJ) in a eolTRPA1+ neuron resting at 35.5 °C. (**c**) Example of a phase-locked response of a caTRPA1-positive neuron resting at 27 °C to a train stimulation at a frequency of 25 Hz . Individual pulse 10 ms, $\sim 1$ mJ. (**d**) Example of a phase-locked response of a caTRPA1+ neuron resting at 27 °C to a train stimulation at a frequency of 50 Hz . Individual pulse 10 ms, $\sim 1$ mJ. (**e**) Expanded inset shown in **d**. Dotted lines are drawn through the APs. (**f**) Example of a phase-locked response of eolTRPA1+ neuron resting at 35.5 °C to a train stimulation at a frequency of 25 Hz . Individual pulse 10 ms, $\sim 1.7$ mJ. (**g**) Expanded inset shown in **f**. Dotted lines are drawn through the APs. (**h**) Control stimulation of a caTRPA1- neuron resting at 27 °C to a train stimulation at a frequency of 50 Hz . Individual pulse 10 ms, $\sim 4.8$ mJ. (**i**) Control stimulation of a eolTRPA1- neuron resting at 35.5 °C to a train stimulation at a frequency of 25 Hz . Individual pulse 10 ms, $\sim 4.8$ mJ.

**Figure 4 | Activation of single cultured neurons expressing caTRPA1.** (**a**) Fluorescent images of neurons co-expressing GGaMP6s and caTRPA1-P2A-tdTomato. The scale bar, 100 µm. (**b**) Stimulation of the TRPA1-expressing neurons using IR irradiation or chemical agonist AITC gives rise to $Ca^{2+}$ transients of comparable amplitudes; $n$ is the number of cells in the group. For each group, averaging was performed over 3–4 stimulation events for each cell in the group of 10–21 cells. ns—not significant. $**P < 0.01$, one-way ANOVA, followed by Bonferroni correction and Duncan *post hoc* test. (**c**) IR stimulation of neuron 1 from (**a**) heated to 25.5 (5.6 mW laser power, black line), 27 (9 mW, red) and 28 °C (11.2 mW, blue). Note different amplitudes of the $Ca^{2+}$ signal at 27 and 28 °C. (**d**) Multiple cycles of neuron 1 activation by mild above-threshold heating using 20 mW laser. Pulse duration is 3 s. (**e**), (**f**) Determination of the neuronal activation threshold with the power of IR laser radiation, focused on neuron 1 (**e**) and neuron 2 (**f**), increased in a stepwise fashion. Localized heating activates only one of two neighbouring cells separated by $\sim 190$ µm. Right $Y$ axes indicate temperature of the neuron 1 (**e**) and neuron 2 (**f**). (**g**–**i**) Activation of caTRPA1-expressing neurons by trains of femtosecond IR laser pulses with a central wavelength of 1,440 nm. The OPO is operated in the pulsed mode to deliver pulse trains (shown by the red line) with a duration $\tau_t \approx 8$ ms (**g**) and 11 ms (**h**) and repetition rate $f_t \approx 1.18$ Hz (**g**) and 0.87 Hz (**h**). Right $Y$ axis delineates average power of millisecond pulse of many femtosecond pulses. (**i**) $Ca^{2+}$ dynamics in caTRPA1-expressing neurons subjected to a single 15-ms train (shown with pink shading) of femtosecond laser pulses with a central wavelength of 1,440 nm. The fluorescence signal is averaged over four heating cycles. Dynamics of GCaMP6s fluorescence is shown in the inset. The error bars represent the s.e.m.

response is even higher than that evoked electrophysiologically in primary afferent zebrafish neurons[51]. It strongly suggests that the dynamics of caTRPA1 channel fits well the speed requirements of a suitable zebrafish thermogenetic activator. Similar result was obtained with 25 Hz/1.7 mJ train stimulation of eolTRPA1+ neurons (Fig. 5f,g), offering a convincing demonstration of fast activation and post-activation recovery of the snake TRPA1 channel. Notably, even trains of pulses with higher $E \approx 4.8$ mJ failed to evoke action potentials in control neurons (Fig. 5h,i). Slower (15 Hz) trains of pulses with $E \approx 27$ mJ at a central wavelength of 1,050 nm also resulted in action potentials locked with pulses to one another (Supplementary Fig. 10e,h).

Electrophysiological measurements performed in parallel with time-resolved studies of $Ca^{2+}$ dynamics in caTRPA1+ neurons have shown that subthreshold laser radiation powers evoking no action potential induce no $Ca^{2+}$ transients either (Supplementary Fig. 12a). Higher laser powers induce both the action potential and build-up of $Ca^{2+}$ Supplementary Fig. 12b). The measured kinetics of $Ca^{2+}$ build-up and decay exhibits an additional delay due to the time lag in the response of the $Ca^{2+}$ indicator relative to the switching between the channel on and off states.

To summarize, our electrophysiological experiments have convincingly demonstrated that the activation of snake TRPA1 channels with IR laser radiation is ideally suited to stimulate neurons in animal kept at physiological temperatures, just a few degrees below the channel activation threshold. caTRPA1 channel also fits well the dynamics of zebrafish afferent neurons. Note however, that using snake TRPA1 to study cultured rodent neurons should be performed at temperatures somewhat lower than normal animal temperature ($\leq 27\,^{\circ}C$ for caTRPA1, $\leq 36\,^{\circ}C$ for eolTRPA1) that might affect neuronal physiology. We therefore advise to use eolTRPA1 in this case, with $T_{ON}$ closer to normal temperature of the mammalian brain. The channel with $T_{ON}$ optimal for *in vivo* stimulation in mammals is yet to be found.

**In vivo thermogenetic stimulation of neurons**. In most of the earlier studies *in vivo*, heat-sensitive TRP channels were activated using either chemical agonists or ambient temperature elevation. These stimulation techniques are inevitably non-local and non-selective, activating all the TRP-expressing cells in a sample. Furthermore, these methods can neither provide repeated activation within a short period of time nor prevent sustained activation of neurons, thus tending to induce toxic effects for the cells because of $Ca^{2+}$ overload. All these issues can be addressed by applying laser radiation for TRP channel activation. In the previous sections, we presented our experiments demonstrating the activation of heterologously expressed caTRPA1 in mouse neurons (Figs 4 and 5). Remarkably, the activation threshold of caTRPA1 in these experiments is just 1° or 2° above 26 °C which is a permissive temperature for *D. rerio* maintenance, enabling *in vivo* activation of caTRPA+ zebrafish neurons using IR laser radiation.

We generated a vector for caTRPA1 expression in zebrafish under control of CREST3 somatosensory enhancer[52] that drives the expression of the channel and tdTomato in trigeminal and Rohon–Beard sensory neurons[53,54] (Fig. 6a,b). The embryos were injected at the single-cell stage and analysed at days 2–2.5 post fertilization. Typically, 5–10 neurons transiently expressing caTRPA1 per embryo were detected as a result of this procedure. Control animals expressed tdTomato only.

Prior to the laser stimulation experiments, we studied the optical transparency of zebrafish larvae at 24 h.p.f. A parallel beam selected from an incandescent lamp with a diaphragm was focused into a 40-μm-diameter spot upon the area of Rohon –

Beard neurons and the transmitted IR radiation was analysed using an IR spectrometer. The IR power loss spectrum measured in this experiment (dash–dotted line in Fig. 6d) closely follows the spectrum of water absorption, as presented in (ref. 35) (dashed line in Fig. 6d) and measured in our experiments for calibration purposes (solid line in Fig. 6d). Stimulation of the fish neurons using laser radiation at 1,440 nm was not efficient due to strong absorption by water in which the larvae were immersed. To identify the most suitable wavelength, we measured the temperature in water as a function of the distance from the bottom of the Petri dish inside the illuminated water column for different wavelengths of IR laser radiation (Fig. 6e). The wavelength of 1,350 nm was chosen as a result of this study, as quasi-cw IR radiation of the femtosecond OPO at this wavelength was found to induce stable temperature profiles within 3 mm from the bottom of the Petri dish, providing enough space for neuron stimulation experiments.

For IR stimulation, the embryos were dechorionated and polymerized in low-melting-point agarose. The embryos were kept at 25 °C for 10–15 min before IR stimulation to let them recover from the stressful embedding procedure. The wavelength-tunable OPO was employed as a source of IR laser pulses for *in vivo* thermogenetic neurostimulation in experiments with zebrafish. A mechanical shutter was used to select isolated 500-ms trains of 100-fs pulses out of the 78-MHz quasi-cw, 1,350-nm OPO output. The IR laser beam was focused with a 25-cm-focal-length lens into an area 60 μm in diameter. Such a broad IR laser beam allowed stimulation of single neurons and small groups of neurons. Laser stimulation of cells induced the escape behaviour in 12 out of the 17 caTRPA1+ embryos studied in our experiments (Supplementary Movie 1, Supplementary Fig. 13). The efficiency of stimulation was found to be a strong function of the laser power. Specifically, laser radiation with $P \approx 12$ mW induced muscle contraction in 23% of the animals. With laser power increased to 18 mW and 30 mW, 72% and 93% of the larvae, respectively, exhibited the escape behaviour (Fig. 6f) with no change in the behaviour of caTRPA1– animals. With a further increase in the pulse power up to 40 mW, escape behaviour was also observed in 11% of caTRPA1– animals. This effect is most probably due to the activation of a higher-$T_{ON}$ endogenous fish TRPV1, which is expressed in somatosensory neurons[55]. Although the dynamic range of the laser powers evoking specific vs. non-specific response is low (from 12 to 40 mW, less than fourfold), this limitation is likely larger due to the intrinsic thermosensitivity of the TRPV1–expressing somatosensory neurons. This emphasizes the importance of choosing a proper channel for thermogenetic activation, which needs to open at a temperature just a few degrees above the normal animal temperature to ensure that only neurons expressing the desired channels are activated with carefully adjusted, very mild heating.

The time lag of the animal response was a function of the power of laser radiation used for stimulation. With $P \approx 30$ mW, the animals started to respond, on average, with a time lag of about $288 \pm 8$ ms (Fig. 6g). Notably, at this laser power, no response from caTRPA1– animals was detected. Given that the channels open, as our electrophysiological studies demonstrate, within 1–3 ms after the temperature reaches the channel activation threshold and that the action potential starts to build-up within another 35–40 ms, the observed time lag of the animal response likely reflects the dynamics of heating.

A typical response of animals to IR stimulation was found to last 0.5–1.1 s, allowing 1–3 tail movements to be captured. The duration of this response showed dependence on the laser intensity (Fig. 6h).

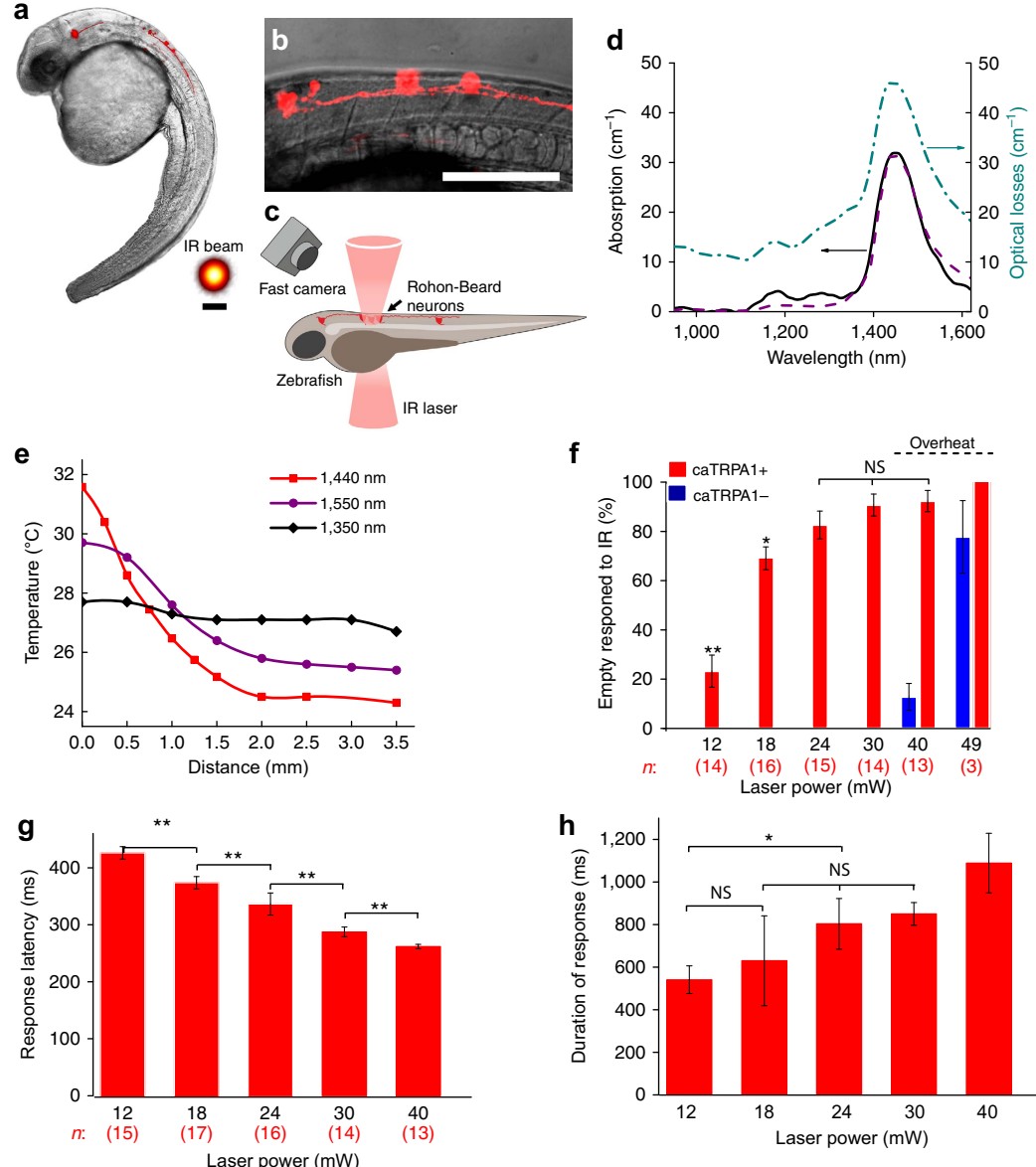

**Figure 6 | *In vivo* activation of caTRPA1-expressing neurons in zebrafish.** (**a**) A fluorescent image of caTRPA1 and tdTomato-expressing neurons overlaid with an image of 2 d.p.f. zebrafish in transmitted light. The inset shows the laser beam; the scale bar, 60 µm. (**b**) A magnified image of Rohon − Beard neurons. (**c**) Diagram of activation of Rohon − Beard neurons and detection of the behavioural response in the fish larva using fast camera. (**d**) The spectrum of water absorption, as measured in a calibration experiment (solid line) and according to (ref. 35) (dashed line) versus the IR attenuation spectrum measured for a 2 d.p.f. zebrafish tissue. (**e**) The temperature measured as a function of the distance from the bottom of the Petri dish with a quasi-cw 35-mW femtosecond OPO output with a central wavelength of 1,350 nm (diamonds), 1,440 nm (squares) and 1,550 nm (circles). (**f**) Escape behaviour of *D. rerio* larvae studied as a function of laser power. Percentage of responded embryos grows up from 12 to 24 mW and shows no significant difference between 24 and 40 mW. At laser powers higher than 40 mW, caTRPA1 − animals start to respond because of the activation of endogenous fish channels; *n* is the number of embryos in experiment. Averaging is over 3 experiments per individual fish. (**g**) The response latency of the escape behaviour as a function of the laser power. *n* is the number of embryos in experiment. Averaging is over five experiments per individual fish. (**h**) Time duration of the animal response to IR stimulation as a function of the laser power. Averaging is over five experiments per individual fish. ns—not significant. *$P < 0.05$, **$P < 0.01$, one-way ANOVA, followed by Bonferroni correction and Duncan *post hoc* test. The error bars represent the s.e.m.

To summarize, the results of *in vivo* experiments demonstrate that IR stimulation is a powerful tool for a thermogenetic activation of living animals, providing spatial and temporal resolution unattainable with agonist-induced stimulation or with ambient temperature changes.

## Discussion

Pit-bearing snake species have evolved thermosensitivity based on the TRPA1 channels expressed in the pit sensory neurons[24].

There are several requirements for such a thermosensing system. First, it should be very sensitive as prey is localized by the snake at a certain distance, up to 1 m. As a result, only a small number of IR photons emitted by the prey reaches the pit membrane. To cope with this, snakes have evolved channels with extremely low activation thresholds. When combined with very high expression levels of TRPA1 in pit neurons, this drastically increases the sensitivity of the pit membrane to low-intensity IR radiation, making sure that the number of open thermosensitive channels in the membrane is sufficient for action potential generation.

Second, the thermosensation system in such snakes should be protected against a sustained open state of its thermosensitive channels. The temperatures many of the snake species are subject to during the day exceed the $T_{ON}$ for their thermosensitive channels, which may lead to $Ca^{2+}$ overload and death of neurons. To prevent this, an efficient mechanism that would shut down the channels sustained in the open state for a long period of time is needed. In our experiments, continuous laser irradiation keeps a substantial fraction of the channels open on the timescale of seconds and even minutes (Figs 3 and 4), resulting in a sustained cytosolic $Ca^{2+}$ increase. As another pertinent result, our studies have revealed that neurons expressing caTRPA1 ($T_{ON} = 28.5\,°C$) can efficiently function even in mammalian cells cultured at a standard cell culture temperature of $37\,°C$, that is, well above the $T_{ON}$ threshold. These findings indicate that some yet unidentified mechanism inactivates the channels kept for a long time at high temperatures. We can only speculate at this stage that this mechanism can involve a partial unfolding of some of the protein substructures under constantly high temperatures. This feature of the caTRPA1 made possible a detailed characterization of the channel response in mammalian neurons presented in our study.

When using ion channels for optogenetic manipulation, the information about correct targeting of the channel to the plasma membrane is important. Usually FPs linked to the channel moiety provide essential information about targeting/mistargeting of the channel. However, sometimes protein termini are essential for the correct protein cellular localization and activity. We therefore tested several possible targeting strategies for caTRPA1 and found that they all, in fact, disturb the channel in some ways. In our experiments, C-terminal fusion of caTRPA1 with FP, although reported earlier as a thermogenetic tool in zebrafish[21], demonstrated very poor performance compared to the N-terminal fusion or wild-type protein. This result can be attributed to the position of C-terminus in the structure of caTRPA1 essential for homotetramer formation[56].

N-terminal fusion mNeonGreen-caTRPA1 was almost as efficient in activating cells as the wild-type channel. However, in neuronal cells, the N-terminal fusion demonstrated a very low expression level (estimated from the low brightness of the fusion in the cells). We therefore used wild-type channels in our experiments.

Heat-sensitive TRP channels can be robustly activated in three different ways. First, chemical agonists such as capsaicin or AITC can activate them, bringing all the advantages and disadvantages of chemogenetics[21]. As a useful feature of this approach, a drug can activate the entire neuronal circuit with a maximal efficiency. Pitfalls include slow drug application/removal times, which excludes fast repeated activation. The second approach is the change of ambient temperature. This approach is used in the most of the reports on thermogenetic manipulation of fruit flies[18,19] and, recently, zebrafish[21]. A change in the temperature in the entire animal allows activation of the entire population of neurons expressing the channel with temporal resolution much higher than that attainable with chemogenetics. However, the spatial resolution is still low, allowing only a certain group of neurons to be activated, providing no means to address individual neurons. The third, optical approach relies on IR laser stimulation of TRP channels. This technique offers numerous advantages, including unprecedented spatial resolution, ultrafast channel activation rates, broadly tunable radiation intensity and wavelength, as well as duration of stimulation that can be easily adapted to a broad variety of organisms. Prior to this work, IR irradiation was used only in a few studies on Drosophila thermogenetics[14,22]. The study describing the FlyMAD approach elegantly demonstrates the flexibility and, to a large extent,

automation of thermogenetic stimulation using IR lasers. The study presented here further develops the idea of using the IR photonics for thermogenetics, demonstrating that this technology enables activation of mouse neurons and zebrafish locomotor responses and showing that the thermogenetic manipulation can be implemented with single-cell resolution and ultrahigh temporal precision.

With rare exception[21], studies involving thermogenetics utilize homologous expression of species-specific TRP channels. In our view, homologous expression should be avoided because the natural host cells can often modify the channel function using numerous post-translation modifications[23]. Heterologously expressed channels, on the other hand, will be less prone to these modifications as many of them are cell- and species-specific. We demonstrate here that, having evolved as molecular tools for remote temperature sensing, snake TRPA1 channels can efficiently depolarize and induce action potentials in mouse and zebrafish neurons stimulated with low-power IR radiation.

While optogenetics with visible light activation provides universal and powerful tools, allowing the same channel to be used in a variety of model organisms, thermogenetics adds yet another important dimension to the technology, offering a diversity of heat-activated TRP channels with distinctly different activation thresholds, making it possible to find the best match for each species. Specifically, caTRPA1 with its $T_{ON}$ of $\sim 28\,°C$ opens at just $1–2°$ above $26–27\,°C$ that falls within the normal zebrafish maintaining temperature range. This channel is thus ideally suited for in vivo studies of neurostimulation in zebrafish, as well as in a broad variety of ectotherm species, such as Drosophila melanogaster and Caenorhabditis elegans. On the other hand, TRP channels with higher activation thresholds, $\sim 38–39\,°C$, would enable in vivo studies on rodent neurons. The eolTRPA1 is a promising tool for such studies. However, accurate measurements of the temperature in the brain of mice and rats are still to be performed.

Yet another important advantage of the thermogenetic technology is that sample cultivation and raising animal for thermogenetic studies does not require artificial disruption of light, which may have unwanted effects on normal circadian rhythms, important for the development of many species, including zebrafish. This suggests the way to avoid one of the serious difficulties of optogenetic technologies, which require larvae expressing photoactivated opsins, for example, to be grown in the dark to prevent unwanted circuit activation.

Heating tissue by $1–3°$ over a short period of time usually does not produce any toxic effect. Stronger heating of TRPA-expressing cells, however, should be avoided as potentially causing cell ablation[21]. Therefore, a careful choice of TRP channels operating just above the animal natural temperature is of crucial importance for thermogenetics. As a reward, mild laser intensities together with high TRP channel conductance and reasonable penetration depths (up to several millimetres[57,58])of IR light in tissues make thermogenetic neurostimulation less invasive compared to canonical optogenetic instruments.

Protecting tissue from overheating in thermogenetic studies engaging IR lasers is as important as protecting neurons from the phototoxicity caused by high-intensity visible light in optogenetics. Local temperatures should be carefully measured in thermogenetic studies to avoid excessive heating, which tends to induce toxic effects and non-specific activation of endogenous TRP channels. In our study, we utilized a fibre-based quantum thermometry, which allows in situ temperature measurements with high spatial resolution and with temperature sensitivity as high as $0.1\,°C$. Such a calibration is critical for an accurate adjustment of the laser power and laser wavelength for a particular organism.

Zebrafish and fruit fly are useful models for optogenetics and thermogenetics. In a recent study[15], wireless magnetic stimulation of deep brain areas in mice using magnetic nanoparticles was demonstrated. In those experiments, a mouse TRPV1 receptor was used as a thermogenetic channel delivered to the brain using viral injection. However, the activation threshold for the TRPV1 channel is about 43 °C (ref. 59). Heating to such a high temperature could be damaging for the tissue. Another foreseeable disadvantage is the post-translational desensitization of TRPV1. In the present study, we demonstrate the utility of the eolTRPA1 channel in the activation of mouse neurons. The activation threshold of this channel is, however, too close to the mouse body temperature and it may be desensitized in in vivo studies. A search for new thermosensitive channels within the TRPA and TRPV subfamilies is needed to identify channels with activation thresholds at ∼40 °C.

To summarize, we have demonstrated, through in vitro and in vivo experiments, a heterologous expression of the snake TRPA1 channels in neurons from different species stimulated by the IR laser radiation with high-precision in situ thermometry using fibre-optic quantum sensors. This combination of technologies has been shown to enable thermogenetic neurostimulation using mild irradiation with the highest spatial and temporal resolution achieved to date in thermogenetic studies.

## Methods

**DNA constructs engineering.** Control unlabelled caTRPA1 (GenBank GU562967) and eolTRPA1 (GenBank GU562966) were cloned to pLenti vector under a cmv promoter with a modified multicloning site (mcs) using BamHI and SalI sites. caTRPA1-mCherry was obtained by moving a linker consisting of 31 amino acids poly-serine-poly-glycine tract (5′-AGTGGTGGTTCAGGTGGTG-GTGGTTCAGGTGGTGGTGGTTCAGGTGGAGGAGGATCAGGAGGAGGAG-GATCAGGAGGAGGATCAGGAGGAGGA-3′) in pLenti (pLVT) vector with modified MCS. First, caTRPA1 was cloned to pLVT with BamHI and SmaI sites. Then, mCherry added to a linker by a two-step PCR reaction was cloned to a full construct using SmaI and SalI sites. mNeonGreen-caTRPA1 was obtained by cloning caTRPA1 to pLVT with SmaI and SalI sites. Then, the mNeonGreen PCR fragment was connected with a 1–60 amino-acid fragment from the N-part of caTRPA1 by overlap extension PCR and cloned to a full construct with BamHI and SmaI sites. caTRPA1-IRES-EGFP and eolTRPA1-IRES-EGFP were generated from pGL3-cmv-IRES2-EGFP vector by cloning TRPA1 genes with XhoI and EcoRI sites. caTRPA1-P2A-tdTomato and eolTRPA1-P2A-tdTomato constructs were generated from pLVT vector. First, we cloned tdTomato added with a P2A sequence by a two-step PCR reaction using SmaI and SalI sites. Then, we inserted TRPA1 genes to this construct with BamHI and SmaI sites for TRPA1₆His containing constructs generation we inserted 6-His epitope (His-His-His-His-His-His) peripheral loop of the channel between D754 and E755 for caTRPA1 and between A754 and T755as it was described earlier for TRPV1 (ref. 16) using overlap extension PCR. Full length TRPA1s were inserted in pLVT containing P2A-tdTomato for caTRPA1₆His using BamHI and SmaI sites and in unlabelled pLVT for eolTRPA6His.

For calcium imaging in eukaryotic cells, GCaMP6s and R-GECO1.1 were cloned to a pCS2 + vector using EcoRI and XbaI sites. For zebrafish vector pC1:CREST3:LexA: LexAop:TRPA1 production, we first cloned a CREST3:LexA:LexAop PCR fragment from pDest:CREST3:LexA:LexAop:CheFtdTomato with NdeI and ApaI to pC1:cmv: HyPer3 construct, which led to a removal of the cmv:HyPer3 element and insertion of a NotI site before ApaI with a reverse primer. Then, caTRPA1 was cloned to a full construct using NotI and ApaI sites. Previous construct pC1:CREST3:LexA:LexAop was used as LexA source in control plasmid mix. pC1:LexAtdTomato for visualization of ca TRPA1 in zebrafish was generated by insertion of PCR fragment LexAop and tdTomato in pC1 vector in triple ligation reaction using NdeI, NaeI and ApaI sites.

For cloning we used BamHI, SmaI, SalI, ApaI, NotI, NdeI, XhoI, NaeI and KpnI produced by NEB and SibEnzyme, Encyclo polymerase (Evrogen), Phusion High Fidelity polymerase (NEB), T4 DNA ligase (Evrogen) and plasmid and DNA extraction kits (Evrogen, Qiagen).

All constructs were verified by sequence (Evrogen).

**HEK293 cell culture and transfection.** HEK293 (ATCC CRL-1573) were authenticated using STR profiling and tested for the absense of mycoplasma contamination using MycoAlert Mycoplasma Detection Kit (Lonza). The cells were cultured in DMEM supplemented with 10% FBS and 20 mM glutamine in a 5% $CO_2$ incubator. The cells were seeded on glass-bottom Petri dishes (in inverted microscopy) or cultured on plastic Petri dishes (in upright microscopy), 50,000

cells/35 mm glass-bottom dish, or 100,000 cells/60 mm for plastic Petri dish. 24 h after planting, the cells were transfected with FuGene HD (Promega) according to the manufacturer's protocol (1 µg µl⁻¹ DNA mix/3 µl of transfection reagent, 2 µg DNA mix/35 mm dish, 4 µg DNA mix/60 mm dish). Then, the cells were cultured in a $CO_2$ incubator for 24 h at 25, 30 or 37 °C to find optimal TRPA1 expression conditions. After our finding that the activation of the cells grown at 25 or 30 °C is low and no toxicity is associated with culturing the cells at 37 °C, all further experiments were carried out with the cells kept at 37 °C after transfection.

**HEK293 thermal and chemical activation.** HEK293 cells were co-transfected with caTRPA1-IRES2-EGFP or eolTRPA1-IRES2-EGFP and $Ca^{2+}$ sensor R-GECO1.1. Thirty-six to forty-eight hours after seeding the cells were transferred to another incubator for temperature conditioning at 25, 30 or 37 °C for 18–26 h. After that half of the dishes were left at the same temperature for additional 1 h and another half were incubated below the TRPA1 temperature threshold (at 25 °C for caTRPA1 or 30 °C for eolTRPA1). Next, cell culture medium was replaced with Hank's balanced salt solution (HBSS) supplemented with 20 mM HEPES and transferred to the stage of the microscope preheated to 25 °C (for caTRPA1) or 30 °C (for eolTRPA1). Cells were activated by the addition of warm HBSS/HEPES under control of electrode thermometer. Final temperature was 30 ± 1 °C (caTRPA1) or 38 ± 1 °C (eolTRPA1). Alternatively, cells were activated using 200 µM AITC in HBSS. For each experimental condition the control cells expressing only R-GECO1.1 were used in the same experimental scheme. As they did not respond to heating or AITC, all the control cells from different temperature regimens were combined into a single control group (Fig. 1b,c).

**Mixed mouse primary embryonic neuronal cell culture.** A mixed mouse primary embryonic neuronal cell culture was prepared as described in ref. 60, with the protocol for animal handling approved by the ethical committee of the Shemyakin − Ovchinnikov Institute of Bioorganic Chemistry.

E17 C57Bl/6 embryos were collected in ice-cold HBSS with low $Mg^{2+}$ and $Ca^{2+}$ (buffer1). Brains were extracted in buffer1, hemispheres were separated from meninges and hippocampi and cortexes collected separately in ice-cold buffer1. The hippocampi and cortexes were incubated in a 0.025% trypsin/EDTA solution for 20 min at 37 °C water bath, washed three times with DMEM (37 °C), supplemented with 10% FBS and 20 mM glutamine, and carefully resuspended with a flame smoothed 1 ml tip for 5–7 times. Cells were plated on poly-D-lysine covered dishes (either plastic 60 or 35 mm glass-bottom dishes, 100 µl of 1 mg ml⁻¹ poly-D-lysine MQ solution per any dish), at 1.5–1.0 × 10⁵ cells in 150 µl of DMEM (37 °C), supplemented with 10% FBS and 20 mM glutamine. 40 min after plating, neurobasal medium supplemented with B27, 5% FBS and 20 mM GlutaMax was added to final volume 2 ml. Once in three days, 50% of neuronal medium was replaced by a fresh neurobasal medium. On the 5th day after plating, the cells were transfected with Lipothectamine LTX with Plus reagent (Thermo Fisher Scientific)) according to the manufacturer's protocol (1 µg µl⁻¹ DNA mix/ 2.5 µl of transfection reagent, 2 µg DNA mix/dish). The mean transfection efficiency was 20 transfected neurons/dish. The co-transfection efficiency is close to 100% (19–20 co-transfected neurons/dish). The cells were kept at $CO_2$ incubator at 5% $CO_2$ and 37 °C for 2 weeks. Note that we observed no toxicity associated with the snake TRPA1 channels expression. Before stimulation experiments the cells were transferred to 25 °C (for caTRPA1) or 30 °C (for eolTRPA1) in Tyrode solution supplemented with 20 mM D-Glucose and 20 mM HEPES pH 7.4. This procedure made the channels active.

**Cell microscopy.** HEK293 and cultured neurons cells were placed into an environmental chamber in HBSS, supplemented with 20 mM HEPES solution at 25, 30 or 37 °C and imaged with a Leica DM6000 wide-field microscope equipped with a 20 × air objective and HC × PL APO lbd.BL 63 × 1.4NA oil objective, T × 2 (tdTomato, R-GECO1.1, mCherry) and GFP (GCaMP6s, EGFP, mNeonGreen) filter cubes. Neuronal cells were imaged 13–14 days after plating in a 2 or 7 ml (35-mm dish or 60-mm dish) of tyrode solution, supplemented with 20 mM D-glucose and 20 mM HEPES, pH 7.4. Time series were analysed using ImageJ software (NIH). All graphs were made in OriginPro 8.6 (OriginLab).

**Data reporting and error analysis.** For imaging experiments, 'n' is a number of independent experiments (individual cells or animals) included in each panels and described in the figure legends. The minimum size of 'n' in the experiments was chosen as 10 and at least three repetitions for every measurement giving us 30 measurements per group. It allows to analyse wide group of data and distribution of effect to ensure adequate power of the statistical test. Statistical analysis was performed testing of with OriginPro 8.6 and SPSS 10.0. For two-groups comparison we used paired (two-tailed) t-test, for multiple comparisons we used one-way ANOVA test followed by Bonferroni correction and Duncan test.

**Laser system for IR photostimulation.** Photostimulation experiments were performed with a laser system (Fig. 2a) consisting of a mode-locked Ti:Sapphire oscillator and a wavelength-tunable femtosecond OPO (ref. 61). The Ti:sapphire oscillator delivered sub-60-fs laser pulses with an energy up to 40 nJ and a central wavelength tunable from 700 to 980 nm at a pulse repetition rate of 78 MHz. The

808-nm output of the Ti:sapphire laser was used as a pump for the OPO, which delivered sub-100-fs pulses with a typical energy of 5 nJ at a pulse repetition rate of 78 MHz, covering the wavelength range from 1,030 to 1,580 nm as its signal-wave output and from 1,680 to 2,100 nm as its idler-wave output. To generate tunable trains of femtosecond pulses for photostimulation experiments, the quasi-cw OPO output was modulated with a mechanical shutter or an electro-optical chopper.

**HEK293 photostimulation.** For HEK293 stimulation, the quasi-cw OPO output at 1,440 nm was loosely focused with a 1-m-focal-length lens into a spot 670 μm in diameter (Fig. 2b), providing a uniform irradiation of a large number of TRPA-expressing cells close to the beam centre. Glass-bottom Petri dishes with cultured cells were placed on a stage of an upright multiphoton microscope (Thorlabs) equipped with XLUMPLFLN objective (NA1.05, Olympus). Continuous-wave laser sources with wavelengths of 473 and 532 nm and an average power up to 50 mW were used for R-GECO1, EGFP, GCaMP6s and tdTomato visualization. Fluorescence from EGFP and GCaMP6s was filtered with an FELH0500 (Thorlabs) low-frequency filter and an FF01-510/42 (Semrock) bandpass filter. The signal from R-GECO1.1 and tdTomato was filtered with a NF533-17 (Thorlabs) notch filter and an FF01-607/70 (Semrock) bandpass filter. Fluorescence images were recorded using a cooled CCD camera 4070M-GE-TE (Thorlabs) with $4 \times 4$ binning and a 900-ms exposure.

**Photostimulation of neurons.** Photostimulation of neurons was studied using the chopped OPO output, delivering trains of femtosecond IR pulses at 1,440 nm with a tunable pulse-train duration and pulse-train repetition rate. The caTRPA1-expressing neurons were activated with isolated 10-ms trains of 1,440-nm femto-second laser pulses produced by optically chopping the OPO output. The IR laser beam was focused with a 40-cm-focal-length lens into an area 60 μm in diameter (See temperature profile on Supplementary Fig. 14). An InGaAs photodiode (PDA10DT, Thorlabs) was used to measure the duration of laser pulse trains.

**Single-cell fibre-optic thermometry.** A fibre-optic temperature sensor used in our experiments integrates an optical fibre, an NV-diamond microcrystal, and a two-wire microwave transmission line[38,39]. To measure the temperature at the chosen site within the cell culture, the microwave field, delivered through the two-wire microwave transmission line integrated with the fibre, is applied to couple the spin sublevels of ground-state NV centres in diamond, polarized by 532-nm laser radiation transmitted through the optical tract of the fibre probe. This laser radiation transfers population from the $^3A$ ground state to the $^3E$ excited state. The photoluminescence (PL) emitted as a result of this process features a zero-phonon line at approximately 637 nm, observed as a well-resolved peak on a broad phonon-sideband line. This PL signal is collected by the same optical fibre. The optical tract of the fibre then serves to transmit this signal to the detection system, which consists of a silicon photodiode, a low-noise preamplifier, and a lock-in amplifier.

Temperature measurements are performed using optically detected magnetic resonance (ODMR)[39,40]. As the temperature of diamond increases, following the temperature in the ambient medium, the profile of the zero-external-magnetic-field ODMR is shifted towards lower microwave frequencies, enabling temperature measurements with a high spatial resolution. For the highest sensitivity and highest speed of local temperature measurements in a cell culture, frequency-modulated microwave spin excitation in NV centres was combined with properly optimized differential lock-in detection[36,37].

**Electrophysiology and imaging of cultured mammalian neurons.** Action potentials in neurons are activated with heat induced by a pulsed output of a compact ytterbium fibre laser (IPG Photonics) , delivering continuous-wave IR radiation at a wavelength of 1,050 nm with an average power up to 10 W, and by millisecond laser pulses at a wavelength of 1,342 nm produced by a diode-pumped solid-state laser. A mechanical shutter (SHB05, Thorlabs) was synchronized with the current-amplification controller (DigiData 1440A ADC board) to select 30- to 50-ms pulses out of the cw output of the ytterbium fibre laser. IR laser radiation was coupled into an optical fibre with a core diameter of 50 μm and a numerical aperture of 0.22 with the help of high-precision translation stages (Thorlabs) and delivered through this fibre to the targeted areas within the neurons under study. The distal end of the fibre was positioned at a distance of 0.3 mm from a targeted cell using a motorized micromanipulator Junior (Luigs and Neumann, Germany). The laser-irradiated area, imaged with a camera, had a shape of an ellipse with principal axes of 90 and 150 μm (Supplementary Fig. 9a).

The temperature $T$ in electrophysiological experiments was determined by measuring the current $I$ through the patch electrode[62],

$$I = I_o \exp\left(-\alpha\left[\frac{1}{T} - \frac{1}{T_o}\right]\right) \qquad (1)$$

where $I_0$ is the initial current at a temperature of 27 or 35.5 °C, set as the initial temperature before the experiment and α is the experimentally determined calibration coefficient. Calibration measurements were performed on the medium that was used for neuron studies. The micropipette electrode was placed inside this medium, which was preheated to 36 °C. We then measured the electric current through this electrode induced by an applied voltage of 22 mV. The best linear fit

for the $I(1/T)$ dependence, plotted using these current readings, yields $\alpha \approx 1{,}311 \pm 57\,K^{-1}$.

The micropipette was then heated by 30- and 50-ms pulses of 1,050-nm laser radiation delivered through an optical fibre whose tip was positioned at a distance of 250 μm from the micropipette, thus mimicking conditions of neuron studies. Supplementary Fig. 9c shows the temperature profiles at the centre of the laser-irradiated area retrieved from current measurements using the running-average algorithm, applied to 100 data points within a 2-ms time window, thus filtering out the high-frequency noise of the electric current. With a pulse energy of 9 mJ, the temperature was found to rapidly increase from 27 to 28.7 °C and 29.3 °C for laser pulse widths of 30 or 50 ms, respectively (Supplementary Fig. 9c), thus heating the medium above the caTRPA1 activation threshold.

A coverslip with transfected neurons was placed in the recording chamber of an upright fluorescence confocal microscope and superfused with bath solution (in mM: 125 NaCl, 25 NaHCO$_3$, 27.5 glucose, 2.5 KCl, 1.25 NaH$_2$PO$_4$, 2 CaCl$_2$ and 1.5 MgCl$_2$, pH 7.4) preaerated with 95% O$_2$, 5% CO$_2$. Patch clamp recording was performed in the current-clamp and voltage-clamp modes. Borosilicate glass electrodes (resistance of 5 MΩ) were filled with a solution containing (in mM) 132 K-gluconate, 20 KCl, 4 Mg-ATP, 0.3 Na$_2$GTP, 10 Na-Phosphocreatine, 10 HEPES, pH 7.25 (all from Sigma, St Louis, MO, USA). The current was increased in a stepwise manner with an Axoclamp 2B amplifier (Axon Instruments, USA) driven with a DigiData 1440A ADC board (both from Molecular Devices, Sunnyvale, CA, USA). During electrophysiological studies, the recording chamber was kept at a temperature of 27 ± 0.5 °C (for caTRPA1 channel) or 35.5 ± 0.5 °C (for eolTRPA1 channel) and perfused at a constant rate of 3 ml min$^{-1}$. Confocal imaging of tdTomato was performed with a Zeiss LSM 5 Live microscope (Germany) in the epifluorescent mode (with excitation at 532 nm and emission at 550 nm).

**Zebrafish egg microinjection and screening.** Animal experiments were performed in accordance with guidelines approved by the Shemyakin-Ovchinnikov Institute of Bioorganic Chemistry (Moscow, Russia) Animal Committee and handled in accordance with the Animals (Scientific Procedures) Act 1986 and Helsinki Declaration.

For zebrafish larvae samples preparation we used protocol based on (ref. 54) with several modifications. 0.58 d needles were filled with 4–7 μl DNA plasmid mix in 0.1 M KCl (experimental mix—pC1:CREST3:LexA:LexAop:caTRPA1 with pC1:LexAop:tdTomato; control mix -pC1:CREST3:LexA:LexAop with pC1:LexAop:tdTomato) and used for injection of 1–2 cell stage embryos into the yolk with calibrated injection volume about 1 nl.

Injected embryos (AB/TL strain) were stored at 26.5 °C in the Petri dish in facility 12:12 light and dark cycle, unfertilized, damaged or dead embryos were periodically removed. [9]. On the second day, post fertilization embryos were manually cleaned from chorion, anaesthetized with 0.02% tricaine and screened for transgene expression using a Leica DM6000 wide-field fluorescent microscope.

Rohon–Beard or trigeminal neuron expression were identified in 40–60% of injected embryos, 5–10 neurons per embryo.

**Zebrafish larvae escape behaviour experiments.** Screen positive embryos were mounted in 1.5% low melt agarose in ddH$_2$O at 35 °C then samples were incubated at 25 °C up to 15 min to decrease high activity of caTRPA$^+$ embryos. Then on a Petri dish agarose was removed from larva's tail with two diagonal cuts at either side of the yolk using a thin razor blade (#20) and pulling agarose away from the trunk and tail of the larva after addition of embryo water.

The wavelength-tunable OPO was employed as a source of IR laser pulses for *in vivo* thermogenetic neurostimulation. A mechanical shutter was used to select isolated 500-ms trains of 100-fs pulses out of the 78-MHz quasi-cw, 1,350-nm OPO output. The IR laser beam was focused with a 25-cm-focal-length lens into an area 60 μm in diameter. The behavioural response was detected using a CCD camera (DCU224M, Thorlabs), operated in the video-recording mode at a rate of 60 fps. To analyse the statistics of the behavioural response, muscle movements of the tail were monitored for full mounted embryos (Supplementary Movie 1). In an experiment with 10 larvae, muscle movements were found to totally correlate with the escape behaviour of larvae.

For larvae used in IR inducible movement analysis the animals were considered as response to IR if tail movement occurred not later than the 100–500 ms IR pulse ends in 10 s long record. For each IR power we produced 2–3 activations for individual embryo. Embryo was counted as responded to IR stimulation only if in all cases of activation it showed a positive response to IR stimulus. In case of negative responses (no tail movement or movement after the pulse) for a given IR power, such larvae were considered as non-responded to IR stimulation.

The response latency was counted as number of frames from the start of IR impulse to the first frame with tail movement and then converted to timescale (ms) using camera predetermined frequency (16.6–60 fps).

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

## Acknowledgements

We thank Y. Panchin for a fruitful discussion, D. Julius for sharing caTRPA1 and eolTRPA1 genes, A. Sagasti for sharing the CREST3- and LexA:LexAop-containing gene constructs, pGP-CMV-GCaMP6s was a gift from Douglas Kim (Addgene plasmid #40753), CMV-R-GECO1.1 was a gift from Robert Campbell (Addgene plasmid #32444). The work was supported by Russian Science Foundation project 14-14-00747 (to V.V.B). The research of E.S.N., V.V.B., A.A.L., I.V.F., A.B.F. and A.M.Z. was supported in part by the Russian Foundation for Basic Research (projects 16-04-00490, 14-29-07263, 14-29-07182, 16-32-80141, 16-52-00190 and 16-29-11799), Molecular and Cell Biology Programme of Russian Academy of Sciences (to E.S.N.), Welch Foundation (Grant No. A-1801) and Government of Russian Federation(project no. 14.Z50.31.0040, Feb. 17, 2017).

## Author contributions

Y.G.E. and A.A.L. contributed equally to this work. V.V.B. and A.M.Z. conceived the project and wrote the text; Y.G.E., A.A.L. and I.V.F. performed the thermostimulation experiments and imaging; I.V.F., A.A.L., D.K. and A.B.F. contributed to the quantum thermometry experiments; D.A.S.-B. provided the expertise in IR laser photonics; I.V.K. contributed to the genetic constructs design and creation; Y.A.B. contributed to the neuronal culture preparation; D.S.B. and A.G.S. maintained and transfected zebrafish; M.R. and E.S.N. performed electrophysiology and imaging; Y.G.E., A.A.L., I.V.F., I.V.K., E.S.N., P.M.B., A.M.Z. and V.V.B. analysed and discussed the data.

## Additional information

**Competing interests:** The authors declare no competing financial interests.

