## [Peer Review File · Nature Communications]

Reviewers' Comments:

Reviewer #1 (Remarks to the Author):

Ermakova and colleagues propose a new technique for stimulating neurons with high spatial and temporal precision (“thermogenetics”). By ectopically expressing thermosensitive protein channels from the heat-seeking pits of different snakes in neurons, the authors are able to elicit calcium influx and action potentials in various in vitro preparations. The authors also use quantum thermometry to measure the corresponding temperature increase during thermal stimulation in order to determine temperature thresholds for these different channels. This suggests that there is potential for a thermogenetic toolbox of various channels that activate at different temperatures that can be used akin to optogenetics. However, the authors fail to demonstrate the technique in vivo in a mammalian model, despite using mammalian neuronal culture for a portion of their experiments. Overall, the paper is very suggestive in demonstrating a novel method of neuronal stimulation, though details regarding laser setups for IR radiation and availability of appropriately channel proteins are unclear. I feel that with extensive revisions (including a rewrite for language), this manuscript would be suitable for publication.

Major Comments

1. The authors tag these thermosensitive proteins using P2A-tomato, however this experiment does not indicate sub cellular localization of the protein. Perhaps, the authors could test to see that at least in cell culture, the channels do localize to the membrane.
2. Figure 1 says that cooling to 26 oC is necessary for thermogenetic activation. However, it is not clear whether this is required for all the preparations (in vitro and in vivo).
3. What is the distance between the neighboring cells in Figure 4f.
4. The control used is the sub-threshold wattage used to produce sub-threshold response in channel-transfected cells. But why not use the action potential-eliciting wattage and show that there is no action potential? It's possible they saw an action potential in control cells so they resorted to the sub-threshold comparison (328-335).

5. The authors evoke action potentials that look entrained to their stimulation frequency. I would suggest using multiple frequencies to show that the response is really time-locked time-locked to the stimulus (342-347).
6. Both 6h and 6g show a trend, but the authors claim 6h is meaningful while 6g is not.
7. If the snake is in ambient temperatures higher than the inactivation threshold of the thermo-sensitive channels, how can they sense heat coming off prey? Does this mean they are sensing heat changes rather than absolute temperatures by threshold? This changes the story quite a bit.
8. If the mNeonGreen fusion was as efficient in activating cells as the wild-type, why does it matter that it had low expression? Isn't that a bonus benefit? Low expression and still activation? Or do they mean that fewer cells were expressing the channel? Needs clarification.
9. "reasonable penetration depths"...what is a reasonable penetration depth?
10. Figure legends are very confusing. No mention of appropriate statistical tests in either the legend or the test. Also, the figures don't include a scale bar for images and the scales are often different within the same figure making it difficult to compare across experiments. Here are some examples:
 - a. Figure 1: Bath temperature is 27C, not mentioned in text, and also higher than the cooling performed in figure 1. Why? No stats? Why not?
 - b. Figure 2: 500um scale bar not labeled in 2a. In 2c, at what ambient temperature is this recording done? Laser power delivers an absolute temperature regardless of the initial temperature of the sample? Maybe I don't understand fully.
 - c. Figure 3: 3a and 3b are difficult to compare because the scales are different, but we want to compare the kinetics of the subsequent stimulations. In 3c and 3d they continue to step up the temperature even after the cells begin to show fluorescence increases...why? Why not hold at the initial temperature that shows a change and see if that temperature eventually builds up that fluorescence? This could significantly alter what they claim as their "threshold".
 - d. Figure 4: "For each group, averaging was performed over 3 to 4 stimulation events". Does this mean per cell in each group? 4a scale bar should read 60µm. 4b is this a multiple comparison? What is being compared to what with the asterisks? Clarify stats. 4c is a more convincing threshold figure than those in 3, but these are mammalian neurons versus HEKs in 3.
 - e. Figure 5: Cells in 5a and 5f were cooled to 27C, not 26C, but this is unclear in the figure. Also the controls in the third line of 5a and 5f would benefit from the action potential evoking control, whereas they use the subthreshold.

f. Figure 6: No scale bar in 6a. Stray error bar in 6b. No stats in figure to show increase in % responders. 6g and 6h could use stats for the claims they make.

Suggestions on writing

I only performed a cursory reading for grammar. I would recommend that the authors consider a manuscript writing service.

line edit

23 Delete second “a”

24 Delete last “A”

28 Delete “of the”

29 Delete “a”

33 Delete “a”

37 Delete “the”

41 Delete “a”

42 Delete first “the”

54 “aiming” should be “aimed”

56 Delete first “the”

62 “which are very likely to induce harmful phytotoxic effects, which are rarely taken into consideration.” should be “which are very likely to induce rarely considered harmful phototoxic effects.

74 Delete last “an”

75 Delete “a”

78 Delete first “the”

83 Delete both “a”

85 Delete first “an”

96 Delete “the”

108 Delete “the” and “an”

109 Delete “the”

113 “functioning” should be “function”

121 Delete “the”

122 Delete “a”

126 “In case” should be “In the case”; delete last “the”

127 Delete “the”; delete “if”

140 “A feature of an IRES” should be “A feature of the IRES”

158 Delete “a”

169 “suggests” should be “offers”

171 Delete second “a”

192 Delete first “the”

200 Delete “a”

202 “out” should be “our”

228 Delete “expressing”

233 Delete “A”

234 Delete first “a”

235 Delete “the”; delete “a”

238 Delete “a”

246 “Several studies demonstrated possibility” should be “Several studies demonstrate the possibility”

248 “neurons” should be “neuron”

251 “knowledge, in fact” should be “measure”

270 Delete “a”

272 Delete “the”

317 “we use the” should be “we used”

318 “electrophysiological technique” should be “electrophysiology”

318 “cultured caTRPA1-P@a-tdTomato transfected hippocampal” should be “cultured hippocampal transfected with caTRPA1”

367 “section” should be “sections”

370 Delete “the”

374 “injected at a single” should be “injected at the single”

379 Delete “analyzed”

380 “IR radiation using an IR” should be “IR radiation was analyzed using an IR”

397 Delete “the”

400 “muscles” should be “muscle”

408 Delete “a”

411 Delete second “the”

427 “requirements for such thermosensing” should be “requirements for such a thermosensing”

428 Delete first “a”

458 Delete “a”; “result” should be “result”

471 Delete first “the”

475 Delete “the”

476 Delete “an”

484 Delete “a”

486 Delete both “a”

491 Delete “the”

506 “do not require” should be “does not require”; delete “an”

507 “effect” should be “effects”

510 “circuits” should be “circuit”

511 Delete “a”

514 Delete “a”

517 Delete “a”

522 Delete “a”

550 “medium” should be “average” (?)

552 Delete “the”

556 “fluorescent” should be “fluorescence”

629 Delete first “the”

Figures Legends

Figures Figures

Reviewer #2 (Remarks to the Author):

The authors report experiments intended to expand the toolbox for thermogenetic activation of neurons. First, they report for the first time the use two TRPA1 channels from the rattle snake to depolarize neurons by light. Second, they have established a setup to illuminate small spots of tissue with infrared light to restrict the activation in space to small parts of tissue or single cells. They discuss their findings in the light of existing thermogenetic approaches.

The work has several very positive aspects. First, the idea to use TRPA1 channels from snakes is innovative and original. One of the channels opens at temperatures above ~ 27.8 °C, the second one at ~ 38.5 °C. The idea to search for more TRPA1 channels in snakes to find the most appropriate ones for specific applications is promising. Second, the experiments on the infrared light stimulation are carefully done and convincingly described. Statistics and controls are appropriate. The conclusions the authors rwa are absolutely valid and convincig.

A somewhat weak point is the fact that the caTRPA1 channel with an opening at 28.5 °C is perhaps useful for experiments on ectotherms, e.g., zebrafish, *C. elegans* or *Drosophila*. However, for experiments on rodents neither of the channels appears appropriate because of the animals' body temperature above the channels' opening threshold. Nevertheless, I believe that the experimental advance made through the description of two snake TRPA1 channels and the technical devices established justifies publication.

Minor points:

1. The TRPA1 channels from diverse species differ considerably in their amino acid sequence and properties (e.g. comparison mouse or snakes vs. *Drosophila*). It should be noted that tagging different channels with fluorophores can have drastically different effects on temperature sensitivity or conductivity. For example, mCherry was fused with *Drosophila* dTRPA1 both at the C- and N-terminus without affecting strongly the temperature sensitivity or conductivity (Vasmer et al. 2014, *Front Behav Neurosci* 8: 174). This should be mentioned to clarify that the negative effects seen with the fluorophore tags in the case of snake TRPA1 is specific for this channel and not a general property of all TRPA1 channels.

2. The authors mention in the Introduction section (line 60) the low conductivity of ChR in order to describe the advantages of thermogenetics. However, there is a large list of different ChR2 variants available, with different conductivities and opening states. For example, ChR2-XXL (Dawydow et al. 2014, *PNAS* 111: 13972) shows a very large conductivity in comparison with wildtype ChR2. This should be mentioned as well.

3. In the Methods section it is not mentioned for how long primary neuronal cultures were kept (several days?) and at what temperature. I assume it was at 37°C. The cell cultures were probably then put to lower temperatures (27°C or 35.5°C) to analyze the effects of the temperature elevation. Please describe that in detail. It should also be mentioned that analyzing rodent neurons at 27 °C or 35.5 °C might affect their physiology. It should be directly stated here that an ideal channel for rodent neurons would open shortly above 38°C (as the authors do in the discussion section). The fact that the channels seem to adapt to 37°C and therefore (if I understood it correctly) can be studied in neuronal cultures is somewhat unsatisfactory. Are there any toxic effects of the TRPA1 expression in the neuronal culture? Do the channels remain open and cause ion fluxes during the time of culturing?

4. The authors mention an in utero electroporation of mouse embryos with caTRPA1. I could not find such an experiment in the manuscript. Are these unpublished experiments, or is there any citation available?

Reviewer #3 (Remarks to the Author):

A) What are the major claims of the paper?

The authors claim to have developed a novel thermogenetic method for the activation of individual neurons which is comparable in performance to certain optogenetic tools with regards to the potency, reversibility and spatiotemporal control.

The authors have expressed snake heat-sensitive TRPA1 channels in HEK293 cell culture, mouse neuronal culture and live zebrafish and characterized their usability for thermogenetic manipulation of neural activity. Using calcium imaging, infrared light, an optical fiber and a spatially precise (~100 um) thermometer they are able to precisely control the temperature of individual neurons in cell culture and establish the temperature thresholds for TRPA1 activation when expressed in HEK293 cells or in cultured mouse neurons. The authors claim that the heat-induced activation of TRPA1 expressing neurons is reversible and very fast (30 ms) and therefore comparable to the kinetics of optogenetic tools. In electrophysiological experiments the authors show that single action potentials can be induced by short stimulations reliably. Finally, the authors expressed rattlesnake TRPA1 in zebrafish sensory neurons and show that escape responses can be elicited reliably within a small window of suitable laser powers.

B) Are they novel and will they be of interest to others in the community and the wider field? If the conclusions are not original, it would be helpful if you could provide relevant references.

The use of TRP channels for heat-induced activation of expressing neurons is not novel (see references 11, 15-20 in the manuscript). The novelty of the presented work lies in the combination of heterologous TRP channel expression with a spatiotemporally precise heat delivery and temperature measurement system. This combination allows rapid and precise control of temperature in individual neurons in cell culture and thus enables optimal spatiotemporal control of TRP channel stimulation.

In my opinion, tools like thermogenetic TRP channels are of high interest to the community and the wider field, since they can enable new experimental strategies. In this specific case, I believe that the presented technique will not be of high interest to the wider field, since its application in the manuscript doesn't offer any obvious advantages over optogenetic approaches (local

illumination and activation of individual neurons) and the achievable control over neural activity is less precise.

The authors state that TRP channels are three orders of magnitude more conductive than the optogenetic channelrhodopsin (line 65), yet in the presented work the needed light intensity for action potential generation is higher than the light intensity needed for channelrhodopsin activation. The use of infrared light can, however, be an advantage, especially when deep tissue penetration is needed or in studies where animals are visually stimulated.

C+E) Is the work convincing, and if not, what further evidence would be required to strengthen the conclusions?

The authors have used a large battery of methods to characterize the usability of caTRPA1 for thermogenetics (precise temporal and spatial control of temperature, different expression systems, calcium imaging, electrophysiology and in vivo validation). However, the physiological data (calcium imaging and electrophysiology), the verification of expression and the achievable spatiotemporal control in vivo is not convincing enough to make TRPA1 a promising tool besides uses in special applications, as explained below:

1) Calcium imaging in HEK cells and neuronal cell culture (Fig. 3+4): The authors claim that TRPA1 induced neuronal activity is fast (onset: 30 ms, offset 3.5 s, Fig. 4g-i), yet Figure 3a-d and Figure 4 c-f show that most of induced transients have much slower kinetics (e.g. decay of activity over the course of a minute in Fig. 4d). Furthermore, the authors claim that TRPA1-induced neural activity is reversible and reproducible (lines 223, 584, 345), however in Fig. 3b the duration of the heat stimulus was increased with every of the two repetitions (i.e. not the same stimulus), in Fig 4d only one repetition is shown and in Fig. 5e only a time window of short ~300 ms is investigated.

2) Electrophysiology (Fig. 5): only the induction of up to 5 action potentials is shown. I would expect (and like to see) that a potent tool for neuronal activation manipulation can be used to induce many more action potentials so that the effect can be measured as firing rate during the heat stimulation period.

3) Expression: The available data does not show whether the TRP channel is exclusively localized to the membrane, as expected.

4) in vivo validation: The authors use a neat set of techniques to enable precise spatial control of temperature at cellular resolution in cell culture (Fig. 2c). However, the spatial resolution of heat stimulation is likely much coarser in vivo due to the three-dimensional, heat-conductive nervous tissue and therefore the advertised high spatiotemporal resolution can likely not be used in vivo.

5) The effects of heat stimulation on TRP expressing cells appear to be i) very variable (e.g. Fig. 4e-f), ii) dependent on the adapted temperatures (Fig.1), and they iii) sometimes show OFF-responses (e.g. Fig. 3a, Supplementary Fig. 3, 80mW) and iv) the neuronal activation kinetics appear to depend strongly on the used laser intensity (Supplementary Fig. 3). This complex response behavior will complicate the application of TRPA1 thermogenetics. Furthermore it has been shown in the past that strong activation of TRP channels can lead to cell ablation (e.g. ref. 19 in the manuscript) and it would be good to see data that this has not been the case in the present study.

D) Validity of any statistical analysis

I did not check the statistical analysis thoroughly, but it appears to be OK.

F+H) On a more subjective note, do you feel that the paper will influence thinking in the field? Please feel free to raise any further questions and concerns about the paper.

In summary, I believe that the presented method is not novel and potent enough to warrant publication of the study in its present form in Nature Communications.

I have a few comments which might help the authors to further improve the manuscript (NOT ordered according to importance):

1. The current manuscript contains many instances of exaggeration without providing quantitative information and I would prefer a more factual style of writing (“arsenal of tools”  “combination of methods”, “ultrahigh-resolution thermometry”  “achieve submillimeter resolution”?, “exceptionally mild, precisely controlled heating”, “far from identifying realistic ways toward deeper penetration”, “optogenetic methods...are the key to understanding functioning of the brain...”, “we tested the above-mentioned procedures”  “we tested some of the above-mentioned procedures”, “dramatically decrease amplitude”  percentage reduction?, “have evolved perfect thermosensitivity”).

2. The optogenetic hyperpolarizer NpHR is a pump, not a channel (line 47)

3. Line 80f: “...evolutionary distant species...lower the risk...”. It is not clear to me why this is to be expected. Also your line 247 appears to contradict this statement.

4. Lines 107-145: This section should be revised, since the negative results in line 108-145 are not very interesting and their presentation can be shortened. The used channel should be the starting point and the positive expression results for the IRES and P2A constructs should be presented here as well. Also a schematic of the different constructs would be helpful.

5. References 27 and 35 are missing

6. Some of the experimental procedures are hard to understand. A) Always state which constructs you use in the different experiments (in some experiments caTRPA1 is used without

any fused fluorophore), e.g. Fig. 1, line 264. B) Indicate which experimental conditions served as control in Fig. 1 and what temperature was used for “no incubation”. An illustrative time line for the temperature changes might help to more easily understand the experiment. If cells were grown at 37°C and then transferred to 30°C without pre-incubation, the temperature step is not “heating” as indicated in the figure legend. C) Indicate the used light powers (or even better – the intensities) in the figure legends (e.g. Fig. 3).

7. Explain AITC when using it for the first time
8. Line 177: why not laser intensity?
9. Line 194: why does a 100fs-pulsed laser reduce the “uncontrollable heat release” as you claim? The heat release of a non-pulsed IR laser can be very easily controlled.
10. Line 282: according to Matthews et al. 2002, the temperature is 28.5°C which would induce TRPA1 activation.
11. Line 549 “Fluorescent response”: the plotted data is a percentage, not fluorescence
12. Line 584 Multiple two
13. Line 585f: what is plotted in e-f? The legend describes one neuron is plotted in each panel e and f, but in the actual figure two neurons are plotted each. What is the difference?
14. Line 718: “@@”?
15. Fig 1b, 25°C, 1h incubation: The error bar is plotted at a percentage of 25%, yet the average percentage appears to be 0%.
16. Fig. 2c: Is the same spatial resolution achieved when 37°C are used as max. temperature?
17. Fig. 3c-d: Please explain how the temperature threshold has been calculated.
18. Fig. 4b: The baseline should be 1, not 0.
19. Fig. 6e: where is the focus of the laser? What is the x/y resolution?
20. Suppl. Fig.2a and legend: The panel 1a indicates “1h incubation at 30°C”, but in the legend 26°C are indicated. Which one is true?

G) References: appropriate credit to previous work

Yes.

We thank the reviewers for the detailed analysis of our paper. The comments helped us to make the manuscript more clear. We have addressed all the reviewers' questions and our step-by-step response can be found below.

Reviewers' comments:

Reviewer #1 (Remarks to the Author):

Ermakova and colleagues propose a new technique for stimulating neurons with high spatial and temporal precision (“thermogenetics”). By ectopically expressing thermosensitive protein channels from the heat-seeking pits of different snakes in neurons, the authors are able to elicit calcium influx and action potentials in various in vitro preparations. The authors also use quantum thermometry to measure the corresponding temperature increase during thermal stimulation in order to determine temperature thresholds for these different channels. This suggests that there is potential for a thermogenetic toolbox of various channels that activate at different temperatures that can be used akin to optogenetics.

Comment

However, the authors fail to demonstrate the technique in vivo in a mammalian model, despite using mammalian neuronal culture for a portion of their experiments.

Response

We would not agree with the term “failed” because we did not try mouse *in vivo* experiments. The main reason for that is that the temperature threshold of eolTRPA1 is too close to normal temperature of the mouse. Therefore it would be too risky to spend animals for these experiments. One of our future aims is to search for TRP channels with the threshold of 39-40 C, just 1-2 degrees above mouse temperature.

Comment

Overall, the paper is very suggestive in demonstrating a novel method of neuronal stimulation, though details regarding laser setups for IR radiation and availability of appropriately channel proteins are unclear. I feel that with extensive revisions (including a rewrite for language), this manuscript would be suitable for publication.

Response

We thank the reviewer for the positive evaluation of our approach to neuronal stimulation.

Comment

1. The authors tag these thermosensitive proteins using P2A-tomato, however this experiment does not indicate sub cellular localization of the protein. Perhaps, the authors could test to see that at least in cell culture, the channels do localize to the membrane.

Response

We performed localization analysis of snake TRPA1 channels with different tags in HEK293 cells and cultured neurons. We were able to check localization for both N- and C-termini FP-tagged versions and for more native version of the channel: the one with 6His epitope inserted into a peripheral loop of the TRPA1. Our results suggest that the channels are targeted mostly to the plasma membrane of both cell types tested with some minor staining of other membranes

within the cells most probably reflecting some protein trafficking routes. The results of the localization experiments are in good agreement with very robust activity of the channels in the mammalian cells.

We have added panels to the Fig. 1 showing HEK293 cells expressing C-terminus tagged version of the caTRPA1 as, in terms of localization, it has an intact N-terminus which is the main determinant of PM targeting and, at therefore reflects localization of non-tagged caTRPA1. Other localization experiments results are summarized in new Supplementary figures 1-3 including FP-tagged and His-tagged channels in HEK293 (Fig S2) and neurons (Fig S3,4).

We have added the following paragraphs and sentences to the corresponding section of the text:

Although caTRPA1-mCherry has somewhat impaired response, it retains native N-terminus which is a main determinant of the channel targeting. Therefore from the localization of the caTRPA1-mCherry we could estimate subcellular distribution pattern of caTRPA1. Figure 1a demonstrates predominantly plasma membrane localization of caTRPA1-mCherry in HEK293 cells indicating proper targeting of the channel. Neuronal expression of caTRPA1-mCherry resulted in mainly plasma membrane targeting of the protein (Supplementary Fig. 3b).

Neuronal expression of dN-mNeonGreen-caTRPA1 resulted in predominant PM targeting of the channel (Supplementary Fig. 3a).

To understand subcellular localization pattern of caTRPA1 we inserted 6-His epitope into a peripheral loop of the channel as it was described earlier for TRPV1⁵ and stained HEK293 cells and neurons expressing caTRPA1_{6His}-P2A-tdTomato with anti-6His antibodies. The channel localized almost exclusively at the plasma membrane of HEK293 cells (Supplementary Fig. 2b) and neurons (Supplementary Fig. 4a,b). However, as 6His-tagged version of the sensor demonstrated slower off kinetics (data not shown) in further experiments we used non-tagged version of caTRPA1.

Similar results were obtained for another snake channel, *Elaphe obsoleta lindheimeri* TRPA1 (eoITRPA1)²³ (Supplementary Fig. S2c, Supplementary Fig. S4c).

Comment.

2. Figure 1 says that cooling to 26 oC is necessary for thermogenetic activation. However, it is not clear whether this is required for all the preparations (in vitro and in vivo).

Response

We used two types of models in our work: *in vitro* (for both caTRPA1 and eoITRPA1) including HEK292 and mouse neurons and *in vivo* (zebrafish, only for caTRPA1). As caTRPA has a temperature threshold at ~28C, there is no need for cooling for *in vivo* experiments because zebrafish is maintained at 26C. For all other experiments *in vitro* we need cooling to make the channels active. We maintain cell cultures expressing the TRPA1 channels at 37C. At this temperature both caTRPA1 and eoITRPA1 localize at the plasma membrane, but the channels remain inactive. There are no signs of toxicity or even embryonic mortality after in utero electroporation. Without cooling to subthreshold temperatures the channels are not sensitive to either heat or AITC. We could only speculate that the channels have some intrinsic mechanisms

(structural rearrangements?) of self-inactivation upon constantly elevated temperatures. Probably the same mechanisms are responsible for protection of snake neurons from calcium overload at high day temperatures.

Comment

3. What is the distance between the neighboring cells in Figure 4f.

Response

The distance between the neurons is ~190 nm. Now the legend for these panels 4ef reads as:

(e), (f) Determination of the neuronal activation threshold with the power of IR laser radiation, focused on neuron 1 (e) and neuron 2 (f), increased in a stepwise fashion. Localized heating activates only one of two neighboring cells separated by ~190 μm .

Comment

4. The control used is the sub-threshold wattage used to produce sub-threshold response in channel-transfected cells. But why not use the action potential-eliciting wattage and show that there is no action potential? It's possible they saw an action potential in control cells so they resorted to the sub-threshold comparison (328-335).

Response

We did not see action potentials from control neurons using any wattages of the laser. We have included corresponding traces as Supplementary figure 10 i,j.

Comment

5. The authors evoke action potentials that look entrained to their stimulation frequency. I would suggest using multiple frequencies to show that the response is really time-locked to the stimulus (342-347).

Response

We performed new stimulation experiments with multiple frequencies (25 Hz, 50 Hz) in addition to 15 Hz trains shown in the first version of the paper. In addition, 25 and 50 Hz trains are longer than in the first version of the ms. All trains of IR pulses induced phase-locked trains of APs. With these new data we rearranged corresponding section of the manuscript, made a new figure 5 and placed most of the data from the old figure 5 to the Supplementary materials.

Comment

6. Both 6h and 6g show a trend, but the authors claim 6h is meaningful while 6g is not.

Response

We agree with the reviewer that both panels show a trend. Now the sentence reads as "The duration of this response showed dependence on the laser intensity (Fig. 6h)".

Comment

7. If the snake is in ambient temperatures higher than the inactivation threshold of the thermo-sensitive channels, how can they sense heat coming off prey? Does this mean they are sensing

heat changes rather than absolute temperatures by threshold? This changes the story quite a bit.

Response

Although this question is out of the scope of the current study, it is very interesting and important, and we speculated many times on possible mechanisms on lab seminars. One explanation could be that the snake uses thermal vision only at late evening-night-early morning. During this time of the day i) the temperature can drop below the threshold and ii) some of the night-active small endotherm animals appear (such as mouse). Therefore the thermal vision can support hunting even in conditions of low light. Another related question: how to avoid overactivation and toxicity of TRPA1 expressing pit neurons at high day temperatures. We could speculate, as mentioned in the paper text, that "some yet unidentified mechanism inactivates the channels kept for a long time at high temperatures. We can only speculate at this stage that this mechanism can involve a partial unfolding of some of the protein substructures under constantly high temperatures". Do the channels sense heat changes rather than absolute temperatures by threshold? The answer is NO, as is we grow the cells at 37C they should be kept at subthreshold temperatures to switch the TRPA1 to the activated state. Probably this process resembles switching ON snake thermal vision at the evening when the ambient air cools down.

Comment

8. If the mNeonGreen fusion was as efficient in activating cells as the wild-type, why does it matter that it had low expression? Isn't that a bonus benefit? Low expression and still activation? Or do they mean that fewer cells were expressing the channel? Needs clarification.

Response

The reason why we decided against using dN-mNeonGreen-TRPA1 is that it has weak fluorescence when expressed in neurons. As a consequence, visualization of transfecter cells is complicated as well as quantification of activated/transfecter neurons ratio. We do not know physical reason for that as single mNeonGreen is 2.5 fold brighter than EGFP.

The effect of low brightness of dN-mNeonGreen-TRPA1 is observed only in neurons, but not in HEK293 cells where the expression level of all the constructs is much higher.

Comment

9. "reasonable penetration depths"...what is a reasonable penetration depth?

Response

Thermogenetics optical toolbox is not linked to absorption spectrum of of some particular opsin, but rather uses laser wavelengths that fit to local minima of radiation attenuation by the tissue to enable activation of he cells within deeper tissue levels. Effective laser penetration depth or transport mean free path, $l_{\text{mfp}} = (\mu_a + \mu_{\text{sc}}(1 - g))^{-1}$, depends on level of tissue scattering anisotropy g , scattering coefficient μ_{sc} and absorption μ_a . For neuronal tissue this depth ranges from 0.7 mm at 473 nm to 3.1 mm at 1290 nm [Jacques, Steven L. "Optical properties of biological tissues: a review." *Physics in medicine and biology* 58.11 (2013): R37].

Now the corresponding sentence reads as: "As a reward, mild laser intensities together with high TRP channel conductance and reasonable penetration depths (up to several millimeters⁵²)

of IR light in tissues make thermogenetic neurostimulation less invasive compared to canonical optogenetic instruments.”

Comment

10. Figure legends are very confusing. No mention of appropriate statistical tests in either the legend or the test.

Response

We have tried to make the figure legends more clear. In particular, we performed additional statistical analysis for figures 1,2,4,6, Supplementary Figure 6. All the statistical tests are described in the corresponding figure legends.

Comment

Also, the figures don't include a scale bar for images and the scales are often different within the same figure making it difficult to compare across experiments. Here are some examples:
a. Figure 1: Bath temperature is 27C, not mentioned in text, and also higher than the cooling performed in figure 1. Why? No stats? Why not?

Response

We have checked the scale bars. Where appropriate, we have made scale bars uniform in all the panels, such as on Fig. S2

We apologize for messy description of the temperature modes used for HEK293 cells activation. We now provide more clear description of the temperatures in the results section and in the MM section where we have added new paragraph on HEK293 stimulation:

“...Then, the cells were cultured in a CO₂ incubator for 24 hours at 25, 30 or 37 °C to find optimal TRPA1 expression conditions. After we found that the activation of the cells grown at 25 or 30 °C is low and no toxicity is associated with culturing the cells at 37°C, all further experiments were carried out with the cells kept at 37 °C after transfection.

HEK293 thermal and chemical activation

HEK293 cells were co-transfected with caTRPA1-IRES2-EGFP or eoITRPA1-IRES2-EGFP) and Ca²⁺ sensor R-GECO1. 36-48 hrs later cells were transferred to another incubator for temperature conditioning at 25, 30 or 37°C for 0.5-4 hrs. Next, cell culture medium was replaced with HBSS supplemented with 20 mM HEPES and transferred to the stage of the microscope preheated to 25 °C (for caTRPA1) or 30 °C (for eoITRPA1). Cells were activated by the addition of warm HBSS/HEPES under control of electrode thermometer. Final temperature was 30 ± 1 °C (caTRPA1) or 38 ± 1 °C (eoITRPA1). Alternatively, cells were activated using 200 μM AITC in HBSS.”

Comment

b. Figure 2: 500um scale bar not labeled in 2a.

Response

Figure 2a is a scheme. The laser beam profile is panel 2b. We marked it more clear now, uniformly with other panels of the figure.

Comment

In 2c, at what ambient temperature is this recording done? Laser power delivers an absolute temperature regardless of the initial temperature of the sample? Maybe I don't understand fully.

Response

IR laser irradiation allows precise controllable heating of the media depending on the laser power (illustrated by Fig 2e for 1440 nm). Resulting medium temperature T depends on the laser driven heating ΔT and an initial sample temperature T_0 . Figure 2c can be an example: initial sample temperature was $T_0 = 19.5$ °C, and heating with 36 mW laser increased the temperature to $T = 28.5$ °C with $\Delta T = 9$. Therefore it is correct to assume that "Laser power delivers an absolute temperature regardless of the initial temperature of the sample", but the resulting temperature will depend on initial temperature of the sample.

We have added basal temperature of the sample to the Fig. 2c legend.

Comment

c. Figure 3: 3a and 3b are difficult to compare because the scales are different, but we want to compare the kinetics of the subsequent stimulations. In 3c and 3d they continue to step up the temperature even after the cells begin to show fluorescence increases...why? Why not hold at the initial temperature that shows a change and see if that temperature eventually builds up that fluorescence? This could significantly alter what they claim as their "threshold".

Response

The panels 3a and 3b of the original Figure 3 reflected different "stages" of the activation process. The panel a better represents a single activation event in HEK293 cells which takes ~20 sec, whereas figure 3b illustrated longer period with multiple activation events. To better compare Ca²⁺ dynamics in HEK293 cells we now have made several new panels, namely Fig 3ef and Supplementary Figure 7c. On the latter figure four sequential activation events are overlaid.

Graphs on the panels 3cd (now 3bc) reflect real experimental data on search of the threshold. As we do not know a priori at which temperature the channel will open, and the temperature calibration takes place after the experiment, we finally exceed temperature threshold for 1-2C, but it does not matter as the channel is already open.

In general, there are two ways of measuring the temperature threshold. Both of them are used in the new version of the paper. On the Figure 3cd (new 3bc) we deliver step temperature jumps (~1°C per step) and define gating temperature T_{ON} as a temperature of rapid Ca²⁺ sensor brightness increase in the cytoplasm. Each temperature step lasts for ~30 sec, enough to open the channel and to induce Ca²⁺ sensor response. Second way suggested by the reviewer was used before by us and others [Gracheva et al "Molecular basis of infrared detection by snakes." Nature 464.7291 (2010): 1006-1011.], [Fedotov, I. V., et al. "Fiber-optic control and thermometry of single-cell thermosensation logic." Scientific reports 5 (2015)]. We have repeated these experiments with caTRPA1 ($n = 34$) and eoTRPA1 ($n = 11$) expressing HEK293 cells using with 1440 nm laser driven heating. Threshold temperature T_a was defined from the

crossing of linear fits of the baseline and steep increase phase. Resulting T_a values, $27.4 \pm 0.4^\circ\text{C}$ and $37.9 \pm 0.8^\circ\text{C}$ for caTRPA1 and eolTRPA1 respectively, are in a good agreement with $T_{\text{ON}} = 27.8 \pm 0.6^\circ\text{C}$ and $38.5 \pm 0.7^\circ\text{C}$. Slightly lower T_a values compared to T_{ON} result from relatively low channel conductance at T_a , not enough to induce Ca^{2+} sensor fluorescence increase. The results are shown in the new Supplementary figure 7a,b.

Comment

d. Figure 4: "For each group, averaging was performed over 3 to 4 stimulation events". Does this mean per cell in each group?

Response

Yes, we averaged 3-4 stimulations per each cell in the group, 5-21 cells in a group. We modified the figure legend accordingly.

Comment

4a scale bar should read $60\mu\text{m}$.

Response

We left only one scale bar on 4a.

Comment

4b is this a multiple comparison? What is being compared to what with the asterisks? Clarify stats.

Response

4b - is a multiple comparison one-way ANOVA for all 6 groups. All TRPA1 expressing groups showed significant differences in mean, and were labeled with asterisks to show level of $p < 0.01$ ($p = 0.0011$). Bonferroni correction was taken to decrease of level of the false-positive result. Control groups were joined in one subgroup by Duncan post-hoc test as they show no significant difference between mean levels, TRPA1 expressing groups cannot be joined in any type of subgroups.

Comment

4c is a more convincing threshold figure than those in 3, but these are mammalian neurons versus HEKs in 3.

Response

We performed similar experiments with HEK293 cells as discussed above.

Comment. Figure 5: Cells in 5a and 5f were cooled to 27°C , not 26°C , but this is unclear in the figure. Also the controls in the third line of 5a and 5f would benefit from the action potential evoking control, whereas they use the subthreshold.

We included resting temperatures to the text. Action potential evoking controls for both channels are included in Supplementary Figure 9j.

Comment

f. Figure 6: No scale bar in 6a.

Response

Now zebrafish embryo image and the inset showing the laser beam shape have the same scale bar 60 μm .

Comment

Stray error bar in 6b.

Response

Stray bars are deleted

Comment

No stats in figure to show increase in % responders.

Response

We have added statistical analysis to the Fig. 6f.

Comment

6g and 6h could use stats for the claims they make.

Response

We have added statistical analysis to the Fig 6g,h.

Comment

Suggestions on writing– ...(multiple)

Response

All corrections are gratefully accepted.

Reviewer #2 (Remarks to the Author):

The authors report experiments intended to expand the toolbox for thermogenetic activation of neurons. First, they report for the first time the use two TRPA1 channels from the rattle snake to depolarize neurons by light. Second, they have established a setup to illuminate small spots of tissue with infrared light to restrict the activation in space to small parts of tissue or single cells. They discuss their findings in the light of existing thermogenetic approaches.

Comment

The work has several very positive aspects. First, the idea to use TRPA1 channels from snakes is innovative and original. One of the channels opens at temperatures above ~ 27.8 °C, the second one at ~ 38.5 °C. The idea to search for more TRPA1 channels in snakes to find the most appropriate ones for specific applications is promising. Second, the experiments on the infrared light stimulation are carefully done and convincingly described. Statistics and controls are appropriate. The conclusions the authors rwa are absolutly valid and convincig.

A somewhat weak point is the fact that the caTRPA1 channel with an opening at 28.5 °C is perhaps useful for experiments on ectotherms, e.g., zebrafish, C-. elegans or Drosophila. However, for experiments on rodents neither of the channels appears appropriate because of the animals' body temperature above the channels' opening threshold. Nevertheless, I believe that the experimental advance made through the description of two snake TRPA1 channels and the technical devices established justifies publication.

Response

We thank reviewer for the positive evaluation of our paper. We think that in fact opening temperature of caTRPA1 of 28.5 is a strong side of this channels enabling very mild stimulation of neurons in ectotherms. If we would have the channel suitable for both ectoterms and endotherms, let say 39-40 C activated, this would mean that we should heat up zebrafish of fly neurons for more than 10-15 C instead of 1-3 C that we have now. This would result in excessive heating and slower ON-OFF transitions because more time would be needed to heat and cool the tissue. Instead, we will continue searching for other channels better suitable for rodent experiments.

Minor points:

Comment

1. The TRPA1 channels from diverse species differ considerably in their amino acid sequence and properties (e.g. comparison mouse or snakes vs. Drosophila). It should be noted that tagging different channels with fluorophores can have drastically different effects on temperature sensitivity or conductivity. For example, mCherry was fused with Drosophila dTRPA1 both at the C- and N-terminus without affecting strongly the temperature sensitivity or conductivity (Vasmer et al. 2014, Front BehavNeurosci 8: 174). This should be mentioned to clarify that the negative effects seen with the flurophore tags in the case of snake TRPA1 is specific for this channel and not a general property of all TRPA1 channels.

Response

We have mentioned the study by Vasmer et al. in the new version of the manuscript. Now the respective parts of the paragraph read as:

In earlier work¹, YFP fusion was reported for human TRPA1, with N-terminus tagging resulting in functional TRPA1, whereas C-terminal fusions displayed significantly impaired activity. Another study reported that both N- and C- tagged mCherry fusions of *Drosophila melanogaster* dTRPA1 retain both temperature threshold and conductivity². We generated *Crotalus atrox* TRPA1 (caTRPA1) chimeras with fluorescent proteins mCherry³ and mNeonGreen⁴ to visualize Ca²⁺ dynamics in HEK293 cells expressing the chimeric constructs using genetically encoded fluorescent Ca²⁺ probes GCaMP6s and R-GECO 1.1. In the case of C-terminal fusions (caTRPA1-mCherry), rapid heating of cells evoked Ca²⁺ response of dramatically decreased amplitude and speed compared to the wild-type channel (Supplementary Fig. 5). Therefore, the

C-terminal fusions of caTRPA1 with FPs are not optimal and should be avoided because of the impaired channel activity.

Comment

2. The authors mention in the Introduction section (line 60) the low conductivity of ChR in order to describe the advantages of thermogenetics. However, there is a large list of different ChR2 variants available, with different conductivities and opening states. For example, ChR2-XXL (Dawydow et al. 2014, PNAS 111: 13972) shows a very large conductivity in comparison with wildtype ChR2. This should be mentioned as well.

Response

We thank the reviewer for bringing our attention to higher conductive ChRs. We included the reference mentioned and made the statement more accurate. Now the sentence reads as follows: "Moreover, despite recent progress in making channelrhodopsins with high single-channel conductance¹⁰ the low conductance of most ChRs (40–60 fS for ChR2)¹¹ necessitates high channel expression levels and high activation light intensities, which are very likely to induce rarely considered harmful phototoxic effects."

Moreover, we should note that ChR2-XXL has very long T_{off} period 76 +/- 12 sec which is much longer compared to snake TRPA1.

Comment

3. In the Methods section it is not mentioned for how long primary neuronal cultures were kept (several days?) and at what temperature. I assume it was at 37°C. The cell cultures were probably then put to lower temperatures (27°C or 35.5°C) to analyze the effects of the temperature elevation. Please describe that in detail. It should also be mentioned that analyzing rodent neurons at 27 °C or 35.5 °C might affect their physiology. It should be directly stated here that an ideal channel for rodent neurons would open shortly above 38°C (as the authors do in the discussion section). The fact that the channels seem to adapt to 37°C and therefore (if I understood it correctly) can be studied in neuronal cultures is somewhat unsatisfactory. Are there any toxic effects of the TRPA1 expression in the neuronal culture? Do the channels remain open and cause ion fluxes during the time of culturing?

Response

We have added the following text to the description of neuronal cultures in the MM section:

"The cells were kept at CO₂ incubator at 5% CO₂ and 37 °C for 2 weeks. Note that we observed no toxicity associated with the snake TRPA1 channels expression. Before stimulation experiments the cells were transferred to 25 °C (for caTRPA1) or 30 °C (for eolTRPA1) in Tyrode solution supplemented with 20 mM D-Glucose and 20 mM HEPES pH 7.4. This procedure made the channels active."

As we mention several times in our paper and in the response to reviews, the snake channels remain closed during the incubation of the cells at 37C, *probably* because of some yet unknown structural mechanism, the same one that protects snake thermosensory pit neurons from Ca²⁺

overload and death during day temperatures exceeding channel T_{ON} . There is no toxicity associated with the channels expression even in the embryo electroporated in utero (personal communication, not shown). As a result, channels in the cells transferred to the microscope stage and imaged immediately from 37C are not active. However, 30-60 min of incubation at subthreshold temperature (25C for caTRPA, 30C for eoTRPA) leads to channel transformation to the active form.

We do not claim that eoTRPA1 can be widely used in mammalian neurons in vivo. However, we clearly demonstrate that it can be used in neuronal cultures at temperatures less than 37C.

As requested, we included the statements about the utility of the channels in mammalian cells. Now the corresponding paragraph reads as the following:

“To summarize, our electrophysiological experiments have convincingly demonstrated that the activation of snake TRPA1 channels with IR laser radiation is ideally suited to stimulate neurons in animal kept at physiological temperatures, just a few degrees below the channel activation threshold. caTRPA1 channel also fits well the dynamics of zebrafish afferent neurons. Note however, that using snake TRPA1 to study cultured rodent neurons should be performed at temperatures somewhat lower than normal animal temperature (≤ 27 °C for caTRPA1, ≤ 36 °C for eoTRPA1) that might affect neuronal physiology. We therefore advise to use eoTRPA1 in this case, with T_{ON} closer to normal temperature of the mammalian brain. The channel with T_{ON} optimal for *in vivo* stimulation in mammals is yet to be found.”

4. The authors mention an in utero electroporation of mouse embryos with caTRPA1. I could not find such an experiment in the manuscript. Are these unpublished experiments, or is there any citation available?

We mentioned *in utero* electroporation to point out that the channel expression is not toxic for mammalian cells and brain even upon expression at the temperatures much higher temperature threshold of the caTRPA1 channel. Possible reason is some intrinsic channel self-protection mechanism making the channel inactive at constantly elevated temperatures. However, as we did not include data from the *in utero* experiments, we removed this statement from the new version of the text.

Reviewer #3 (Remarks to the Author):

A) What are the major claims of the paper?

The authors claim to have developed a novel thermogenetic method for the activation of individual neurons which is comparable in performance to certain optogenetic tools with regards to the potency, reversibility and spatiotemporal control.

The authors have expressed snake heat-sensitive TRPA1 channels in HEK293 cell culture, mouse neuronal culture and live zebrafish and characterized their usability for thermogenetic manipulation of neural activity. Using calcium imaging, infrared light, an optical fiber and a spatially precise (~100 um) thermometer they are able to precisely control the temperature of individual neurons in cell culture and establish the temperature thresholds for TRPA1 activation when expressed in HEK293 cells or in cultured mouse neurons. The authors claim that the

heat-induced activation of TRPA1 expressing neurons is reversible and very fast (30 ms) and therefore comparable to the kinetics of optogenetic tools. In electrophysiological experiments the authors show that single action potentials can be induced by short stimulations reliably. Finally, the authors expressed rattlesnake TRPA1 in zebrafish sensory neurons and show that escape responses can be elicited reliably within a small window of suitable laser powers.

B) Are they novel and will they be of interest to others in the community and the wider field? If the conclusions are not original, it would be helpful if you could provide relevant references. The use of TRP channels for heat-induced activation of expressing neurons is not novel (see references 11, 15-20 in the manuscript). The novelty of the presented work lies in the combination of heterologous TRP channel expression with a spatiotemporally precise heat delivery and temperature measurement system. This combination allows rapid and precise control of temperature in individual neurons in cell culture and thus enables optimal spatiotemporal control of TRP channel stimulation.

Comment

In my opinion, tools like thermogenetic TRP channels are of high interest to the community and the wider field, since they can enable new experimental strategies. In this specific case, I believe that the presented technique will not be of high interest to the wider field, since its application in the manuscript doesn't offer any obvious advantages over optogenetic approaches (local illumination and activation of individual neurons) and the achievable control over neural activity is less precise.

Response

Whereas local illumination and individual neurons are both achievable using ChR family of tools, thermogenetics provides advantages of IR light excitation which is not visible for small models such as fly and fish and does not bring phototoxicity which is almost unavoidable in traditional optogenetics. The present study adds for the first time cellular resolution to thermogenetic activation of neurons making this technique now a real alternative to optogenetics.

Comment

The authors state that TRP channels are three orders of magnitude more conductive than the optogenetic channelrhodopsin (line 65), yet in the presented work the needed light intensity for action potential generation is higher than the light intensity needed for channelrhodopsin activation. The use of infrared light can, however, be an advantage, especially when deep tissue penetration is needed or in studies where animals are visually stimulated.

Response

We completely agree with the point that IR light brings advantages of deep penetration and absence of side-effects due to activation of animal visual pathway. But the most positive aspect of using IR instead of visible wavelengths is much less phototoxicity. There are many intrinsic fluorophores in the cells that absorb visible wavelengths: flavins, hemes, and others. They all, in presence of oxygen, produce reactive oxygen species (ROS) upon illumination due to the photodynamic effect. The more light we use, the more ROS we produce. The advantage of IR is that cellular fluorophores do not absorb it directly and the energy of individual photons is too low to induce phototoxicity. Therefore, even if the nominal laser power is higher than in some ChR experiments, we do not induce oxidative damage of tissue. Rather we precisely heat small local volume of water to change temperature. Taken together, direct comparison of laser powers

between visible and IR ranges does not give reliable information because different physical modes of interaction of these wavelengths with the living matter.

Comment

The authors have used a large battery of methods to characterize the usability of caTRPA1 for thermogenetics (precise temporal and spatial control of temperature, different expression systems, calcium imaging, electrophysiology and in vivo validation). However, the physiological data (calcium imaging and electrophysiology), the verification of expression and the achievable spatiotemporal control in vivo is not convincing enough to make TRPA1 a promising tool besides uses in special applications, as explained below:

1) Calcium imaging in HEK cells and neuronal cell culture (Fig. 3+4): The authors claim that TRPA1 induced neuronal activity is fast (onset: 30 ms, offset 3.5 s, Fig. 4g-i), yet Figure 3a-d and Figure 4 c-f show that most of induced transients have much slower kinetics (e.g. decay of activity over the course of a minute in Fig. 4d). Furthermore, the authors claim that TRPA1-induced neural activity is reversible and reproducible (lines 223, 584, 345), however in Fig. 3b the duration of the heat stimulus was increased with every of the two repetitions (i.e. not the same stimulus), in Fig 4d only one repetition is shown and in Fig. 5e only a time window of short ~300 ms is investigated.

Response

We thank the reviewer for this comment as when preparing the first version of the paper we missed these important points. We now add the results of additional experiments to the ms.

First, we should note that the results of calcium imaging report well only the first (ON) phase of the response because i) the Ca²⁺ influx is fast and ii) the calcium probe responds quickly. However, OFF phase of Ca²⁺ transient is slow as i) Ca²⁺ should dissociate from the probe and ii) Ca²⁺ concentration in the cytoplasm decays slowly depending on the activity of calcium pumps and exchangers. Therefore Ca²⁺ imaging is useful, but not the most precise way to track the channel opening-closing dynamics. This is even better illustrated on the panel Supplementary Fig Nb where electrophysiology and Ca²⁺ imaging are performed simultaneously. You can see that when the channel is already closed Ca²⁺ is still being pumped out slowly. In case of Figures 3 a-d and 4 c-f we stimulate the cells within long periods of heating (tens of seconds), therefore we expect large concentrations of Ca²⁺ in the cytoplasm and then slow decay. In contrast, when the stimulation is done by short pulses of the laser (Fig4g-i) calcium concentration within the cells can not reach such high values and Ca²⁺ pumps remove calcium ions from the cytoplasm much faster.

We agree that the panel 3b is not convincing and now we replaced it with the graph illustrating four sequential activation in HEK293 cells events using the heat stimulation of the same duration.

The same with the panel 4d. We now replaced this panel with the graph of 5 sequential stimulation events in the neurons using the same stimulation duration.

Both these new data sets illustrate our initial claim that the channel can be repeatedly activated, but now in much more clear way.

Figure 5e we discuss in the response to the next comment.

Comment

2) Electrophysiology (Fig. 5): only the induction of up to 5 action potentials is shown. I would expect (and like to see) that a potent tool for neuronal activation manipulation can be used to induce many more action potentials so that the effect can be measured as firing rate during the heat stimulation period

Response

We performed new stimulation experiments with multiple frequencies (25 Hz, 50 Hz) in addition to 15 Hz trains shown in the first version of the paper. In addition, 25 and 50 Hz trains are longer than in the first version of the ms. All trains of IR pulses induced phase-locked trains of APs. With these new data we rearranged corresponding section of the manuscript, made a new figure 5 and placed most of the data from the old figure 5 to the Supplementary materials.

Comment

3) Expression: The available data does not show whether the TRP channel is exclusively localized to the membrane, as expected.

Response

We performed localization analysis of snake TRPA1 channels with different tags in HEK293 cells and cultured neurons. We were able to check localization for both N- and C-termini FP-tagged versions and for more native version of the channel: the one with 6His epitope inserted into a peripheral loop of the TRPA1. Our results suggest that the channels are targeted mostly to the plasma membrane of both cell types tested with some minor staining of other membranes within the cells most probably reflecting some protein trafficking routes. The results of the localization experiments are in good agreement with very robust activity of the channels in the mammalian cells.

We have added panels to the Fig. 1 showing HEK293 cells expressing C-terminus tagged version of the caTRPA1 as, in terms of localization, it has an intact N-terminus which is the main determinant of PM targeting and therefore reflects localization of non-tagged caTRPA1. Other localization experiments results are summarized in new Supplementary figures 1-4 including FP-tagged and His-tagged channels in HEK293 (Fig S2) and neurons (Fig S3,4).

We have added the following paragraphs and sentences to the corresponding section of the text:

Although caTRPA1-mCherry has somewhat impaired response, it retains native N-terminus which is a main determinant of the channel targeting. Therefore from the localization of the caTRPA1-mCherry we could estimate subcellular distribution pattern of caTRPA1. Figure 1a demonstrates predominantly plasma membrane localization of caTRPA1-mCherry in HEK293 cells indicating proper targeting of the channel. Neuronal expression of caTRPA1-mCherry resulted in mainly plasma membrane targeting of the protein (Supplementary Fig. 3b).

Neuronal expression of dN-mNeonGreen-caTRPA1 resulted in predominant PM targeting of the channel (Supplementary Fig. 3a).

To understand subcellular localization pattern of caTRPA1 we inserted 6-His epitope into a peripheral loop of the channel as it was described earlier for TRPV1⁵ and stained HEK293 cells

and neurons expressing caTRPA1_{6His}-P2A-tdTomato with anti-6His antibodies. The channel localized almost exclusively at the plasma membrane of HEK293 cells (Supplementary Fig. 2b) and neurons (Supplementary Fig. 4a,b). However, as 6His-tagged version of the sensor demonstrated slower off kinetics (data not shown) in further experiments we used non-tagged version of caTRPA1.

Similar results were obtained for another snake channel, *Elaphe obsoleta lindheimeri* TRPA1 (eolTRPA1)²³ (Supplementary Fig. S2c, Supplementary Fig. S4c).

Comment

4) *in vivo* validation: The authors use a neat set of techniques to enable precise spatial control of temperature at cellular resolution in cell culture (Fig. 2c). However, the spatial resolution of heat stimulation is likely much coarser *in vivo* due to the three-dimensional, heat-conductive nervous tissue and therefore the advertised high spatiotemporal resolution can likely not be used *in vivo*.

Response

We do not think that moving from 2D to 3D models we would lose demonstrated high spatial resolution of the stimulation. Moreover, in 3D tissue additional dimension allows faster heat dissipation after activation improving temporal resolution of the activation. Heat conductivity of the nervous tissue can be described by heat conductivity constant a^2 which differs only for 10% for brain ($0.13 \text{ mm}^2 \text{ s}^{-1}$) and water ($0.143 \text{ mm}^2 \text{ s}^{-1}$). Therefore heat conductivity of the nervous tissue is also not a limiting factor for spatio-temporal resolution of thermogenetics *in vivo* compared to *in vitro*.

We have performed additional measurement of the temperature profiling of a water volume heated up with 18 mW 1440 nm laser of 60 μm beam width (Supplementary fig. 14). The results demonstrate locality of heating within $\sim 130 \mu\text{m}$ area. This area is much more restricted compared to the one on the Fig. 2b (wide beam to activate large group of the cells). In case of heating using pulsed laser (neuronal and fish experiments) the heating volume should be even more narrow.

Comment

5) The effects of heat stimulation on TRP expressing cells appear to be i) very variable (e.g. Fig. 4e-f), ii) dependent on the adapted temperatures (Fig.1), and they iii) sometimes show OFF-responses (e.g. Fig. 3a, Supplementary Fig. 3, 80mW) and iv) the neuronal activation kinetics appear to depend strongly on the used laser intensity (Supplementary Fig. 3). This complex response behavior will complicate the application of TRPA1 thermogenetics. Furthermore it has been shown in the past that strong activation of TRP channels can lead to cell ablation (e.g. ref. 19 in the manuscript) and it would be good to see data that this has not been the case in the present study.

Response

- i) The main component of the response of neurons on Fig 4f is Ca²⁺ elevation in the cytoplasm at the temperature exceeding threshold. The same can be seen at the figures 3b,c, 4c,e,f. Kinetics of the channels may vary depending on heating intensity (OLD Supplementary Fig.3). Regarding variability visible at Fig. 4ef, note

that heating here is delivered using long irradiation steps 20 sec long each. This condition leads to complex Ca²⁺ load/efflux and spiking behavior in the neurons. The aim of these experiments was to demonstrate that the channels have predictive T threshold and we can activate one neighbouring cell without activating another. When we use more precise modes of activation, like pulsed IR laser source or shorter periods of stimulation, we immediately achieve more stable less variable response (Fig 4gh, 5c-g)

- ii) Indeed, cells should be kept for 30-60 min at subthreshold temperature if expressed at 37C, because the channels demonstrate unique insensitivity to either heat or AITC when expressed at constant above-threshold temperature. However this allows working with mammalian cells without toxicity associated with constantly open conformation of the channel.
- iii) We suggest that OFF-responses visible at some graphs (such as 3a, OLD Supplementary Fig3) reflect rapid change of the refractive index of the medium because of media temperature change as no single electrophysiological measurement using patch clamp revealed additional current after switching the laser off.

We have to note that OLD Supplementary Figure 3 shows activation of HEK239 cells, not neurons. But we agree with the reviewer that kinetics of activation depends on the laser power that could be caused by different proportion of activated channels at temperatures close to threshold. Note that ChR current also depends strongly on the light intensity. We rather think that this dependency is an advantage as it allows more careful and precise control over cell activation.

Comment

I have a few comments which might help the authors to further improve the manuscript (NOT ordered according to importance):

1. The current manuscript contains many instances of exaggeration without providing quantitative information and I would prefer a more factual style of writing (“arsenal of tools”  “combination of methods”, “ultrahigh-resolution thermometry”  “achieve submillimeter resolution”?, “exceptionally mild, precisely controlled heating”, “far from identifying realistic ways toward deeper penetration”, “optogenetic methods...are the key to understanding functioning of the brain...”, “we tested the above-mentioned procedures”  “we tested some of the above-mentioned procedures”, “dramatically decrease amplitude”  percentage reduction?, “have evolved perfect thermosensitivity”).

Response

We have corrected the majority of suggested phrases. However we left “exceptionally mild, precisely controlled heating” as we really achieve exceptional control over heating, left we tested the above-mentioned procedures as now, in the current version of the paper, we tested them all, and left “have evolved perfect thermosensitivity” because the thermosensitivity of the pit snakes is really perfect in our opinion.

Comment

2. The optogenetic hyperpolarizer NpHR is a pump, not a channel (line 47)

Response

Thanks for the note. "Channels" is changed to "pumps" in the text.

Comment

3. Line 80f.: "...evolutionary distant species...lower the risk...". It is not clear to me why this is to be expected. Also your line 247 appears to contradict this statement.

Response

TRP channels are complex and could be post-translationally regulated by phosphorylation, protein-protein interactions etc. However such modifications and interactions are species-specific. Kinases not only recognize phosphorylation sites, but they need also docking site to be present in protein. Therefore it is unlikely that docking and phosphorylation sites for mammalian or fish kinases will be present in snakes. The same is true for multiple possible protein-protein interactions that regulate channel function. As an example to make this idea more clear: when we developed HyPer, a genetically encoded fluorescent sensor for H₂O₂, we made it out of two parts: E.coli OxyR protein and jellyfish GFP. When we express this protein in mammalian cells and measure the diffusion rate by FRAP, we see that the sensor diffuses in the cell extremely quickly meaning that it likely has no interaction partners within mammalian cells. In contrast, it is well known that using Ca²⁺ probes based on mammalian calmodulin in mammals brings unwanted phenotypes because of interactions of this calmodulin with CaM-binding proteins. We therefore believe that cross-species transfer of protein blocks is more advantageous than using host-specific activators or sensors.

Line 247 was confusing as it was written in present tense. We change it to the past tense: However, whether non-mammalian TRPA channels can be used for mammalian neuron activation remained unclear.

Comment

4. Lines 107-145: This section should be revised, since the negative results in line 108-145 are not very interesting and their presentation can be shortened. The used channel should be the starting point and the positive expression results for the IRES and P2A constructs should be presented here as well.

Response

We shortened the section leaving only the intro paragraph and the data on non-fused channels expression. As the rest of the section contains an important information on fused channels and a new information about subcellular localization of the channels, and this part mostly refers to new Supplementary figures, we moved this part to the Supplementary Note 1.

Comment

Also a schematic of the different constructs would be helpful.

Response

We have made a new Supplementary figure 1 with schemes of all the constructs used in our work.

Comment

5. References 27 and 35 are missing

Response

Corrected

Comment

6. Some of the experimental procedures are hard to understand. A) Always state which constructs you use in the different experiments (in some experiments caTRPA1 is used without any fused fluorophore), e.g. Fig. 1, line 264.

Response

We mention the constructs used in the new version of the paper.

Comment

B) Indicate which experimental conditions served as control in Fig. 1

Response

We used cells transfected only with Ca²⁺ indicator. We have added this statement to the figure legend.

Comment

and what temperature was used for “no incubation”.

Response

“No incubation” means that cells were transferred onto the microscope stage directly from 37°C and stimulated without preconditioning. Now it is more clear indicated in the M&M section.

Comment

An illustrative time line for the temperature changes might help to more easily understand the experiment. If cells were grown at 37°C and then transferred to 30°C without pre-incubation, the temperature step is not “heating” as indicated in the figure legend.

Response

As the reviewer 1 also found our description of HEK stimulation experiments not clear enough, we have changed a Fig. 1 legend and have added the corresponding section to the M&M part of the manuscript. Regarding particular condition mentioned by the reviewer, cells were preconditioned for 1 hr at 25 °C before stimulation.

Comment

C) Indicate the used light powers (or even better – the intensities) in the figure legends (e.g. Fig. 3).

Response

We have added now the energy of heating light to all the figure legends where IR is used.

Comment

7. Explain AITC when using it for the first time

Response

Now the sentence reads as "...temperature conditioning within an hour is needed to make the cells respond to either heat or TRPA1 agonist allyl isothiocyanate (AITC)."

Comment

8. Line 177: why not laser intensity?

Response

The meaning of this statement, as reflected in the following text, is that using proper wavelengths we achieve efficient heating of the tissue while avoiding light absorption by cellular intrinsic fluorophores that causes phototoxic effects. However we agree with the reviewer that at any given wavelength varying light intensity allows to adjust heating degree.

Comment

9. Line 194: why does a 100fs-pulsed laser reduce the "uncontrollable heat release" as you claim? The heat release of a non-pulsed IR laser can be very easily controlled.

Response

We agree with the reviewer and corrected the claim. Now the sentence reads as "The phototoxic effects are further reduced^{33,34} by using ultrashort (≈ 100 fs) IR pulses for laser irradiation."

Comment

10. Line 282: according to Matthews et al. 2002, the temperature is 28.5°C which would induce TRPA1 activation.

Response

Zebrafish can be maintained at different temperatures. In our fish facility, similar to many other ones, fish is maintained at 26 °C. The reviewer mentioned the reference where the fish maintenance temperature was 28.5 °C

Comment

11. Line 549 "Fluorescent response": the plotted data is a percentage, not fluorescence

Response

Thanks, we have corrected this in the figure legend.

Comment

12. Line 584 Multiple two

Response

We have changed this panel with one showing 5 sequential events

Comment

13. Line 585f: what is plotted in e-f? The legend describes one neuron is plotted in each panel e and f, but in the actual figure two neurons are plotted each. What is the difference?

Response

Panels 4e-f show GCaMP6s signal for neuron 1 (black line) and neuron2 (blue line). The difference is that on the panel 4e we focus heating laser on the neuron1 (and it responds faster) and on 4f we heat up the neuron 2 (and it responds first). Accordingly, right Y axes show temperature of neuron 1 (4e) and 2 (4f) (added to the figure legend).

Comment

14. Line 718: "@@"?

Response

Now the sentence reads as "...laser pulses with an energy up to 60 nJ and a central wavelength tunable from 700 to 980 nm at a pulse repetition rate of 78 MHz."

Comment

15. Fig 1b, 25°C, 1h incubation: The error bar is plotted at a percentage of 25%, yet the average percentage appears to be 0%.

Response

The main bar was missing, now corrected.

Comment

16. Fig. 2c: Is the same spatial resolution achieved when 37°C are used as max. temperature?

Response

Spatial distribution of heat depends on the area of heating by the external source and on thermal conductivity of the matter. Therefore spatial resolution will not be affected with the increase of heating intensity until the entire volume of the media will not be profoundly heated.

Comment

17. Fig. 3c-d: Please explain how the temperature threshold has been calculated.

Response

We repeat there the response to similar question of the reviewer 1.

In general, there are two ways of measuring the temperature threshold. Both of them are used in the new version of the paper. On the Figure 3cd (new 3bc) we deliver step temperature jumps (~1°C per step) and define gating temperature T_{ON} as a temperature of rapid Ca^{2+} sensor brightness increase in the cytoplasm. Each temperature step lasts for ~30 sec, enough to open the channel and to induce Ca^{2+} sensor response. Second way suggested by the reviewer was used before by us and others [Gracheva et al "Molecular basis of infrared detection by snakes." Nature 464.7291 (2010): 1006-1011.], [Fedotov, I. V., et al. "Fiber-optic control and thermometry of single-cell thermosensation logic." Scientific reports 5 (2015)]. We have repeated these experiments with caTRPA1 ($n = 34$) and eoTRPA1 ($n = 11$) expressing HEK293

cells using with 1440 nm laser driven heating. Threshold temperature T_a was defined from the crossing of linear fits of the baseline and steep increase phase. Resulting T_a values, $27.4 \pm 0.4^\circ\text{C}$ and $37.9 \pm 0.8^\circ\text{C}$ for caTRPA1 and eolTRPA1 respectively, are in a good agreement with $T_{\text{ON}} = 27.8 \pm 0.6^\circ\text{C}$ and $38.5 \pm 0.7^\circ\text{C}$. Slightly lower T_a values compared to T_{ON} result from relatively low channel conductance at T_a , not enough to induce Ca^{2+} sensor fluorescence increase. The results are shown in the new Supplementary figure 7a,b.

Comment

18. Fig. 4b: The baseline should be 1, not 0.

Response

Corrected.

Comment

19. Fig. 6e: where is the focus of the laser? What is the x/y resolution?

Response

The laser in the fish experiments was focused 1 mm from the surface of the Petri dish. This is a distance at which the fish larva normally positioned after embedding into the 1% low melting agarose. However, in case of 1350 nm laser precise positioning along the optical axis is not critical as IR beam has confocality parameter of 4.2 mm. This defines almost uniform radiation intensity near the focus. Spatial resolution of the heating measurement in our system was defined by the size of the diamond ($30 \mu\text{m}$) and x/y width of the heating volume was $130 \mu\text{m}$ (Supplementary figure 14)

Comment

20. Suppl. Fig.2a and legend: The panel 1a indicates "1h incubation at 30°C ", but in the legend 26°C are indicated. Which one is true?

Response

Incubation was done at 30°C for eolTRPA1.

Reviewers' Comments:

Reviewer #1 (Remarks to the Author):

The authors have addressed all of my concerns. However, they could add a couple of sentences in the discussion about broad applicability. In particular, address the concern that temperature changes might affect the targeted tissue non-specifically.

Reviewer #2 (Remarks to the Author):

The authors have spent considerable effort to improve the manuscript and to clarify the reviewers' questions/comments. In particular, in the new version of the manuscript the authors

- demonstrate the plasma membrane localization of the channels more convincingly;
- clarified the time of incubation of the cells at distinct temperatures;
- added convincing experiments with multiple stimulation frequencies to demonstrate that action potentials are time-locked with stimulation pulses;
- added indications of statistical tests;

I have only one minor comment on citing references for the application of thermogenetics in *Drosophila*. Reference 21 is not appropriate in the context of ambient temperature changes, because in this study IR laser light focused on the gustatory organ was used. Also, reference 18 is not a "typical" example for the use of dTRPA1 in *Drosophila*. More appropriate citations would be, e.g., Pulver et al., *J Neurophysiol.* 2009 Jun;101(6):3075-88 or Vasmer et al., *Front Behav Neurosci.* 2014 May 15;8:174.

Overall, the authors have addressed all critical points satisfactorily. I think the manuscript contains novel and interesting information that advances the field of thermogenetics, even if the tools might perhaps not be broadly applied. I recommend publication of the manuscript.

Reviewer #3 (Remarks to the Author):

In their 2nd submission the authors have significantly improved the experiments and descriptions of the methods and have addressed many of my concerns. Taken together, the study is an interesting piece of work that explores the application of heat-sensitive channels for local neural activity manipulation. I believe this exploration will be of high interest to the neural circuit analysis community. However, the presented tools themselves come with limitations, which will

likely be a burden for applying the presented methods in other research projects. The quality of the experiments is adequate for publication in Nature Communications, however the presentation and discussion could still be improved.

Major points:

1. Lines 215 -221: As other reviewers have suggested, the employed multi-step protocol for temperatures isn't very well suited for dissociating the temperature threshold (of interest here) from time-dependent or cumulative effects. A reported temperature-threshold for the TRPA1 channels would be much more convincing if it was shown as DC experiment (just a single step).

2. Usefulness for the neural circuit analysis community is questionable, since a lot of parameters need to be precisely controlled (see a-f below). It will still be interesting for special applications where visible light needs to be avoided, but I'd prefer if each of these potential limitations were labeled as such in the manuscript (not ordered according to importance):

a. Small dynamic range of laser powers before starting to affect non-expressing neurons with the laser (12- 40 mW, i.e. less than one order-of-magnitude, see lines 414-417).

b. "local temperature should be carefully measured in thermogenetic studies to avoid excessive heating" (line 530)  technological limitation for in vivo studies.

c. Kinetics of activation is dependent on the applied laser power (Fig. S7d), which is different from the situation of tools like channelrhodopsin.

d. Prolonged stimulation leads to prolonged post-stimulation neural activity or calcium levels (Fig. S7d, Fig. 4c).

e. Sometimes OFF-responses occur after turning the illumination off. The authors argue in their response that "...OFF-responses ...reflect rapid change of the refractive index of the medium because of media temperature change as no single electrophysiological measurement using patch clamp revealed additional current after switching the laser off". I don't think this explanation is likely to be the correct one. In Fig. S7d the OFF-responses are visible only for the two highest stimulation intensities. According to (Bashkatov and Genina, Proceedings of SPIE 2003) the refractive index of water only changes by 0.1% when shifting the temperature from 20°C to 30°C.

f. Limited spatial resolution of heating: In Fig S14, heating is still evident ~300 um away from the stimulation site (23° instead of the basal 19.5° C).

3. In Figure 1, a schematic of the experimental protocol (incubation, temperature steps) should be given, so that the reader has a chance to understand the gist of the experiment by looking at the figure alone.

Minor points:

Line 46: realistic  effective?

Line 202: "...radial distribution...is fully controlled by the radial field intensity profile...". The language should be toned down here. There is no evidence that other factors do not play a role, i.e. it has not been shown that a smaller fiber diameter would lead to the expected smaller heating spot. Suggestion "fully controlled"  "corresponds to"

Line 254: "comparable to the size of a single cell"? This does not seem to be correct or it is unclear what you mean. The size of the cell soma is about 20 μm in the picture, three times smaller than the laser beam diameter.

Line 275: "...the basal temperature of zebrafish maintenance in fish facilities is 26-27°C". Please provide two or three references for this claim. I could only find preferred temperatures of around 28°C in the literature. This point is quite important given the authors' claim that the fish facility temperature is sub-threshold for the channel.

Line 285-287: please provide a better specification of the parameters (pulse of pulses, where was the power measured? At the sample? What power are you reporting? Time-averaged power? Average power during a millisecond pulse of femtosecond pulses? Peak power during the femtosecond pulse?

Line 343 and 353: Why did you use 27 mJ for evoking 15Hz pulses and 1 mJ for 25 or 50 Hz pulses (factor ~20)? Was this necessary to make this experiment work? If so, why?

Line 388: transfected  transiently expressing

Line 399: "...Petri dish for different..."  "...Petri dish inside the illuminated water column for different..."

Line 430: "muscle contraction initiation": No, the hundreds of millisecond delay are not very

likely to be explainable by slow initiation of muscle contraction. E.g. during escape responses animals respond within less than 10 ms.

Line 495: “with single-cell resolution”  In my opinion, single cell resolution has not been shown. The cells in Figure 4 were quite distant to each other and in Fig. 4e both cells are activated although only one cell has been targeted.

Line 503: “...TRPA1 channels can perform with equal robustness...” What does the word robustness mean here? Please specify.

Fig 4e: Why were both neurons activated if only one neuron was targeted? This should be mentioned/discussed in the manuscript.

Fig. S12 and S7d: the authors explain in their response to my comments that “you can see that when the channel is already closed Ca^{2+} is still being pumped out slowly”, arguing that the slow off kinetics in their treatment is due to the slow calcium dynamics of the neuron. However, it seems also possible that a fraction of TRPA1 channels stays open and only closes slowly. In other calcium imaging studies that I am acquainted with, even after strong activation the calcium signal goes back to baseline within a few seconds.

The possibility that TRPA1 activation leads to cell ablation (as shown in ref. 19 of the old ms for TRP channels) should be discussed.

Line 186: “...the phototoxic effects are further reduced by using ultrashort ... IR pulses...”: It is unclear to me why this should reduce phototoxicity. If a CW laser was used (same average power), the phototoxicity should be much lower and the (necessary) heating effect should be the same!

Response to reviewers

We thank the reviewers for generally positive evaluation of our efforts to improve the manuscript. Please find below our point-by-point response to the comments.

Reviewer #1 (Remarks to the Author):

Comment

The authors have addressed all of my concerns. However, they could add a couple of sentences in the discussion about broad applicability. In particular, address the concern that temperature changes might affect the targeted tissue non-specifically.

Response

This comment is somewhat similar to the more specific comment on potential cell ablation by the reviewer #3. Excessive heating can induce cell death. We have added a couple of rather obvious words to the discussion:

"Heating tissue by 1 to 3 degrees over a short period of time usually does not produce any toxic effect. Stronger heating, however, should be avoided **as potentially causing cell ablation.**"

Reviewer #2 (Remarks to the Author):

The authors have spent considerable effort to improve the manuscript and to clarify the reviewers' questions/comments. In particular, in the new version of the manuscript the authors

- demonstrate the plasma membrane localization of the channels more convincingly;
- clarified the time of incubation of the cells at distinct temperatures;
- added convincing experiments with multiple stimulation frequencies to demonstrate that action potentials are time-locked with stimulation pulses;
- added indications of statistical tests;

Comment

I have only one minor comment on citing references for the application of thermogenetics in *Drosophila*. Reference 21 is not appropriate in the context of ambient temperature changes, because in this study IR laser light focused on the gustatory organ was used. Also, reference 18 is not a "typical" example for the use of dTRPA1 in *Drosophila*. More appropriate citations would be, e.g., Pulver et al., *J Neurophysiol.* 2009 Jun;101(6):3075-88 or Vasmer et al., *Front Behav Neurosci.* 2014 May 15;8:174.

Response

We have corrected the references.

Reviewer #3 (Remarks to the Author):

In their 2nd submission the authors have significantly improved the experiments and descriptions of the methods and have addressed many of my concerns. Taken together, the study is an interesting piece of work that explores the application of heat-sensitive channels for local neural activity manipulation. I believe this exploration will be of high interest to the neural circuit analysis community. However, the presented tools themselves come with limitations, which will likely be a burden for applying the presented methods in other research projects. The quality of the experiments is adequate for publication in Nature Communications, however the presentation and discussion could still be improved.

Major points:

Comment

1. Lines 215 -221: As other reviewers have suggested, the employed multi-step protocol for temperatures isn't very well suited for dissociating the temperature threshold (of interest here) from time-dependent or cumulative effects. A reported temperature-threshold for the TRPA1 channels would be much more convincing if it was shown as DC experiment (just a single step).

Response

We apologize that we did not write it clear enough while responding to the comment of the previous review round. While performing the experiments on temperature threshold determination with HEK cells using steps of laser activation, we started each step from the basal temperature 21°C. The results were presented on the Fig. S7a and S7b.

We have now changed the corresponding sentence in the figure legend accordingly: "... Heating at each point was achieved by 20 sec irradiation of 1440 nm laser, **each time** starting from the basal temperature 21 °C..."

Therefore we have already responded to this comment.

Comment

2. Usefulness for the neural circuit analysis community is questionable, since a lot of parameters need to be precisely controlled (see a-f below). It will still be interesting for special applications where visible light needs to be avoided, but I'd prefer if each of these potential limitations were labeled as such in the manuscript (not ordered according to importance):

a. Small dynamic range of laser powers before starting to affect non-expressing neurons with the laser (12- 40 mW, i.e. less than one order-of-magnitude, see lines 414-417).

Response

We agree with the reviewer that the dynamic range of the laser powers is quite small. However, potential limitations largely depend on the biological system used. In the experiments with *Daino rerio* the control neurons are intrinsically thermosensitive: zebrafish somatosensory neurons sense temperature along with other parameters and they express endogenous TRPV1. The key point here is that caTRPA1 enable selective laser-evoked depolarization of these neurons and we are able (yet with low dynamic range) to activate these neurons with the temperature ramps much smaller than those needed to activate neurons in the control fish. When we work with the

non-heat-sensing neurons, such as mammalian cortical and hippocampal neurons, depolarization of the control neurons is more than 10-fold smaller compared to the *ca-* or *eo*TRPA1-expressing neurons responding to the same amount of the laser radiation. We have added the sentence on that to the corresponding paragraph of the Results section. Now the end of the paragraph reads as follows: "Although the dynamic range of the laser powers evoking specific vs. non-specific response is low (from 12 to 40 mW, less than 4-fold), this limitation is largely due to the intrinsic thermosensitivity of the TRPV1-expressing somatosensory neurons. This emphasizes the importance of choosing a proper channel for thermogenetic activation, which needs to open at a temperature just a few degrees above the normal animal temperature to ensure that only neurons expressing the desired channels are activated with carefully adjusted, very mild heating."

Comment

b. "local temperature should be carefully measured in thermogenetic studies to avoid excessive heating" (line 530)  technological limitation for in vivo studies.

Response

Currently existing technologies [1-3] do not allow in vivo brain temperature measurements with desired spatial resolution and accuracy. However we don't think that this is a serious limitation for using thermogenetics in neuroscience. There are examples of magnetic-induced thermogenetics in the mouse where temperature increase in the vicinity of the injected magnetic particles. Instead, we use here a calibration using microdiamonds and we continuously develop updates of these methods and computer algorithms for quantification of tissue heat conductivity patterns [Refs. 38-39 from the paper]. Moreover, as the laser powers used in classical optogenetics are usually high enough to heat the tissue, we would recommend to take temperature effects into account in optogenetics as well.

[1] Kiyatkin, Eugene A., P. Leon Brown, and Roy A. Wise. "Brain temperature fluctuation: a reflection of functional neural activation." *European Journal of Neuroscience* 16.1 (2002): 164-168.

[2] Kozak, L. R., Bango, M., Szabo, M., Rudas, G., Vidnyanszky, Z., & Nagy, Z "Using diffusion MRI for measuring the temperature of cerebrospinal fluid within the lateral ventricles." *Acta Paediatrica* 99.2 (2010): 237-243.

[3] DeBow, Suzanne, and Frederick Colbourne. "Brain temperature measurement and regulation in awake and freely moving rodents." *Methods* 30.2 (2003): 167-171.

Comment

c. Kinetics of activation is dependent on the applied laser power (Fig. S7d), which is different from the situation of tools like channelrhodopsin.

Response

We agree with the reviewer that kinetics of TRPA1 activation depends on the laser power as reported on Fig. S7d. However, we could not agree that this situation is different from the optogenetics based on channelrhodopsins. Studies on ChRs photoactivation demonstrate that photoinduced current amplitude and speed of its increase depend on activating light intensity. One of the examples is given below:

Lin, J.Y., et al., *Characterization of engineered channelrhodopsin variants with improved properties and kinetics*. *Biophys J*, 2009. **96**(5): p. 1803-14

The panels B1-B3, C1 of the figure 2 of this paper demonstrate electrophysiological responses of HEK293 cells expressing three rhodopsins, ChR2, ChEF, and ChIEF to various intensities of the activating light. It is clearly demonstrated that the activation speed and amplitude depends on the light density similar to TRPA1 in our experiments (Figure S7d).

We therefore cannot state that this is a technology limitation different from opsins.

Spectral and kinetic properties of ChR variants to varying light density and duration. (A) Spectral responses of ChR2 (A1), ChEF (A2), and ChIEF (A3). The vertical lines indicate the estimated peaks. All responses normalized to the maximum response obtained from the cell tested at the various wavelengths ($n = 5$ for ChR2, ChEF; $n = 6$ for ChIEF). (B) Examples of ChR2 (B1), ChEF (B2), and ChIEF (B3) responses to 0.11, 0.48, 2.59, 9.64, and 19.81 mW/mm² of light provided by an LED 470 nm light source. Note the faster channel closure after light removal for ChIEF compared to ChEF. (C1) The intensity-amplitude and intensity-onset (C3) relationship of ChR2 (black, $n = 8$), ChEF (light gray, $n = 7$), and ChIEF (dark gray, $n = 11$) for the maximum response (C1) and the plateau component of the response (C2) normalized to projected maximum response of the individual cell tested. Introduction of I170V (ChIEF) reduced the EC₅₀ of ChEF by 2.3× (for the maximum response) and 3× (for the plateau response). (D) Responses of ChR2 (D1), ChEF (D2) and ChIEF (D3) to 1, 2, 3, 4, 5, 10, and 20 ms of light stimulation at ~19.8 mW/mm². From Lin, J.Y., et al., *Characterization of engineered channelrhodopsin variants with improved properties and kinetics*. Biophys J, 2009. **96**(5): p. 1803-14

The panels B1-B3, C1 of the figure 2 of this paper demonstrate electrophysiological responses of HEK293 cells expressing three rhodopsins, ChR2, ChEF, and ChIEF to various intensities of the activating light. It is clearly demonstrated that the activation speed and amplitude depends on the light density similar to TRPA1 in our experiments (Figure S7d).

We therefore cannot state that this is a technology limitation different from opsins.

Comment

d. Prolonged stimulation leads to prolonged post-stimulation neural activity or calcium levels (Fig. S7d, Fig. 4c).

Response

Indeed, in situations of prolonged stimulation with laser pulses longer than 1 s we can see prolonged activity (Fig.4c– 4f, Fig.S7d). While we do not know the reason for this shape of the signal, we could not call this behavior "limitation" as it can be easily bypassed using pulsed heating. We have demonstrated (Figs. 4g, 4h) that long (~40 sec) heating of the neuron with IR pulses of $\tau_t = 8$ ms is reflected in Ca^{2+} sensor signal bursts with T_{off} same as GCaMP6s $\tau_{1/2} \approx 3$ s (Fig. 4i) [ref 33]. This indicates that there is no prolonged neuronal activity after each pulse and after the entire set of the pulses. Moreover, our electrophysiological recordings confirm these observations with even better resolution, and at the level of channel activity recordings (Fig 5). Generation of action potentials correlate only with the laser pulses in a phase-locked manner.

Comment

e. Sometimes OFF-responses occur after turning the illumination off. The authors argue in their response that "...OFF-responses ...reflect rapid change of the refractive index of the medium because of media temperature change as no single electrophysiological measurement using patch clamp revealed additional current after switching the laser off". I don't think this explanation is likely to be the correct one. In Fig. S7d the OFF-responses are visible only for the two highest stimulation intensities. According to (Bashkatov and Genina, Proceedings of SPIE 2003) the refractive index of water only changes by 0.1% when shifting the temperature from 20°C to 30°C.

Response

Indeed, continuous heating of the cells for 10-15 °C we observe post-stimulatory Ca^{2+} burst. Together with temperature-induced optical aberrations there could be temperature-dependent changes in IP_3R channel and SERCA pump [refs 46,47] leading to release of Ca^{2+} from the ER stores. However, this is not a limitation of our technology, as this problem does exist only at conditions of relative overheating and does not exist when careful (2-4 °C) heating and pulsed irradiation are used. And, again, let us repeat our previous claim: "...no single electrophysiological measurement using patch clamp revealed additional current after switching the laser off." Current reflects real behavior of the channel, whereas Ca^{2+} dynamics reflects behavior of complex network of Ca^{2+} channels and pumps.

Comment

f. Limited spatial resolution of heating: In Fig S14, heating is still evident ~300 um away from the stimulation site (23° instead of the basal 19.5° C).

Response

Similar to optogenetics, thermogenetics has spatial limits determined by the spatial distribution of laser radiation and by the length of the pulse τ_t . We have demonstrated that we can use pulses as short as $\tau_t = 8$ ms for stimulation. This restricts size of the heat distribution area to a few tens of micrometers. Working with cell cultures we were able to reach single-cell resolution of the heating. Nevertheless, focusing IR radiation to highly dispersing deep brain tissue we could expect dispersion if the laser light. Therefore, efficiency of deep tissue stimulation will depend on correct choice of wavelength, focusing geometry and use of the pulsed irradiation. Selectivity of the activation will depend on depth of the area of interest and optical properties of the tissue.

Comment

3. In Figure 1, a schematic of the experimental protocol (incubation, temperature steps) should be given, so that the reader has a chance to understand the gist of the experiment by looking at the figure alone.

Response

We have added the scheme to the Fig. 1

Minor points:**Comment**

Line 46: realistic effective?

Response

Agreed, changed.

Comment

Line 202: "...radial distribution...is fully controlled by the radial field intensity profile...". The language should be toned down here. There is no evidence that other factors do not play a role, i.e. it has not been shown that a smaller fiber diameter would lead to the expected smaller heating spot. Suggestion "fully controlled"  "corresponds to"

Response

Agreed, changed as suggested.

Comment

Line 254: “comparable to the size of a single cell”? This does not seem to be correct or it is unclear what you mean. The size of the cell soma is about 20 μm in the picture, three times smaller than the laser beam diameter.

Response

The full sentence cited by the reviewer is the following: " In our experiments with IR laser radiation, the diameter of the laser beam ($\approx 60 \mu\text{m}$) was comparable to the size of a single cell, thus enabling activation of individual neurons without affecting the neighboring cells (Fig. 4 a,e,f)."

The size of the cell body, $\sim 20 \mu\text{m}$, is comparable with the diameter of the beam ($\approx 60 \mu\text{m}$), not even an order of magnitude difference. As we have a deal with neurons, the effective size of the cell is larger than the size of the cell body, because axons and dendrites also have TRPA1 and contribute to stimulation of the neuron.

Comment

Line 275: "...the basal temperature of zebrafish maintenance in fish facilities is 26-27°C". Please provide two or three references for this claim. I could only find preferred temperatures of around 28° C in the literature. This point is quite important given the authors' claim that the fish facility temperature is sub-threshold for the channel.

Response

Temperature range used for zebrafish cultivation is wide. Various sources recommend different ranges. Our facility maintains the fish at 26-27 °C which is within optimal range (26-28.5 C) according to the Ref 48 in the text: «The room temperature or the tank temperature is generally maintained between 26-28.5 °C».

Another example:

«Temperature: The document states on p. 24, “A water temperature of 28.5° C is widely cited as the optimum temperature for breeding zebrafish. There has however, been little research to investigate the full implications of constantly keeping fish at this very specific temperature.” While the document recommends a very specific temperature of 28.5° C as the optimum for breeding zebrafish, it provides a number of references with a range of 25-29° C and acknowledges that more research is needed to determine temperature preference and implication of maintaining fish at warmer temperatures for an extended period of time. Performance standards should be applied with consideration of health, welfare and species-typical behavior.” From Reed B, Jennings M (2011) Guidance on the housing and care of Zebrafish. Research Animals Department, Science Group, RSPCA 62

One more example:

«The thermal preference or optimum for zebrafish has not been formally defined. The maintenance temperature of 28.5 °C recommended by Westerfield (1995) is almost universally cited for zebrafish in culture. This optimum is supported experimentally by at least one published study, in which zebrafish showed a marked increase in growth when held at $28 \pm 0 \text{ }^\circ\text{C}$ as opposed to lower and/or stochastically and predictably fluctuating temperatures (Schaefer and Ryan, 2006). However, based upon their wide range of tolerance in nature, it is highly probable that their actual preference range extends considerably above and below this

temperature, and the wider range of 24–30 °C recommended by Matthews et al. (2002)³ is probably more appropriate» From the Ref 50 in the text.

Moreover, similar to neurons and HEK cells, zebrafish expressing caTRPA1 can be maintained at temperatures higher than the channel threshold because of TRPA1 auto-protection mechanism an resistance to slow heating or constantly elevated temperature. They need to be cooled down to 26-27 °C before the experiment.

We have changed the sentence and provided the references:

"Since the basal temperature of zebrafish maintenance in fish facilities is generally maintained between 26-28.5°C⁴⁸⁻⁵⁰, these measurements prove that TRPA1 channels offer a powerful tool to study *Danio rerio* neurophysiology."

Comment

Line 285-287: please provide a better specification of the parameters (pulse of pulses, where was the power measured? At the sample? What power are you reporting? Time-averaged power? Average power during a millisecond pulse of femtosecond pulses? Peak power during the femtosecond pulse?

Response

We report average power during a millisecond pulse of femtosecond pulses. The power was measured at the sample. We have added the following sentence to the Fig. 4 legend: " Right Y axis delineates average power of millisecond pulse of many femtosecond pulses."

Comment

Line 343 and 353: Why did you use 27 mJ for evoking 15 Hz pulses and 1 mJ for 25 or 50 Hz pulses (factor ~20)? Was this necessary to make this experiment work? If so, why?

Response

As it was mentioned in the text, different wavelengths were used: 1 mJ refers to 1342 nm, 27 mJ refers to 1050 nm.

We performed experiments with various wavelengths (1050, 1032, 1440 nm) in order to demonstrate how important choosing proper parameters of the stimulating light. The ability to adjust central wavelength allows working with different biological systems. 1440 nm radiation falls within the local maximum of water absorption ($\alpha_{1440} = 32 \text{ cm}^{-1}$) and therefore ideally suited for cell activation through the bottom of the Petri dish under inverted microscope, as demonstrated in the experiments with cultured neurons and HEK293 cells. At 1350 nm water absorption drops to ($\alpha_{1350} = 3 \text{ cm}^{-1}$) allowing to deliver heating radiation through the water volume to the sample when focusing through the lens or optical fiber. Therefore we have chosen this wavelength in the experiments with *D. rerio* and with electrophysiology. 1050 nm absorption by water is even less ($\alpha_{1050} = 0.18 \text{ cm}^{-1}$) and could be potentially used in future for deep brain stimulation.

Importantly, we were able to use all these wavelengths in our system. However, for each of them we have to adjust power to achieve the same level of heating.

Comment

Line 388: transfected  transiently expressing

Response

Agreed, changed as suggested.

Comment

Line 399: "...Petri dish for different..."  "...Petri dish inside the illuminated water column for different..."

Response

Agreed, changed as suggested.

Comment

Line 430: "muscle contraction initiation": No, the hundreds of millisecond delay are not very likely to be explainable by slow initiation of muscle contraction. E.g. during escape responses animals respond within less than 10 ms.

Response

We agree with the reviewer that this interpretation is very speculative. We removed that and now the corresponding sentence reads as "Given that the channels open, as our electrophysiological studies demonstrate, within 1 – 3 ms after the temperature reaches the channel activation threshold and that the action potential starts to build up within another 35 – 40 ms, the observed time lag of the animal response reflects the dynamics of heating."

Comment

Line 495: "with single-cell resolution"  In my opinion, single cell resolution has not been shown. The cells in Figure 4 were quite distant to each other and in Fig. 4e both cells are activated although only one cell has been targeted.

Response

As can be seen on the panels 4e and 4f, when we focus the light on the Neuron 1 in the time range 60-80 sec we have selective activation of this neuron, but not the Neuron 2. Similarly, when we activate the neuron 2 on the panel 4f, in the time range 90-120 sec we do not see activation of the Neuron 1. The neurons are positioned at the distance of contact by processes with each other.

Comment

Line 503: "...TRPA1 channels can perform with equal robustness..." What does the word robustness mean here? Please specify.

Response

We think that "equal robustness" possibly has some quantification meaning. We have changed the sentence to the following:

"We demonstrate here that, having evolved as molecular tools for remote temperature sensing, snake TRPA1 channels can efficiently depolarize and induce action potentials in mouse and zebrafish neurons stimulated with low-power IR radiation."

Comment

Fig 4e: Why were both neurons activated if only one neuron was targeted? This should be mentioned/discussed in the manuscript.

Response

This happens only when we use continuous irradiation. Heat redistribution in the medium during strong continuous heating leads to activation of the neighboring cells. This heat flow leads to differences between the shapes of temperature distribution and activating IR radiation as demonstrated on Fig S14. Solution of this problem is using pulsed irradiation allowing much more local heating.

We have added a corresponding sentence: In our experiments with IR laser radiation, the diameter of the laser beam ($\approx 60 \mu\text{m}$) was comparable to the size of a single cell, thus enabling activation of individual neurons without affecting the neighboring cells (Fig. 4 a,e,f). **Further heating leads to activation of the neighboring neuron (Fig. 4 e,f).**

Comment

Fig. S12 and S7d: the authors explain in their response to my comments that "you can see that when the channel is already closed Ca^{2+} is still being pumped out slowly", arguing that the slow off kinetics in their treatment is due to the slow calcium dynamics of the neuron. However, it seems also possible that a fraction of TRPA1 channels stays open and only closes slowly. In other calcium imaging studies that I am acquainted with, even after strong activation the calcium signal goes back to baseline within a few seconds.

Response

We agree that one of the explanations could be slower channel closing dynamics upon continuous irradiation. Another reason could be temperature effects on SERCA and IP3R (Refs 46,47). However, this issue can be solved by using pulsed radiation, as we demonstrate in the manuscript.

Comment

The possibility that TRPA1 activation leads to cell ablation (as shown in ref. 19 of the old ms for TRP channels) should be discussed.

Response

Well defined temperature threshold of the channel, choosing the channel with proper threshold for each biological model with the activation temperature in the physiological range, and control of the heating intensity and temperature calibration ensures that the chances of ablation are not higher than in other optical methods such as confocal microscopy or two-photon neuronal stimulation.

We have added the corresponding text in the Discussion section. Now it reads as: "Heating tissue by 1 to 3 degrees over a short period of time usually does not produce any toxic effect. Stronger heating, however, should be avoided **as potentially causing cell ablation.**"

Comment

Line 186: "...the phototoxic effects are further reduced by using ultrashort ... IR pulses...": It is unclear to me why this should reduce phototoxicity. If a CW laser was used (same average power), the phototoxicity should be much lower and the (necessary) heating effect should be the same!

Response

We agree with the reviewer and removed the claim on phototoxicity from the sentence. Now the corresponding text reads as following: " For highest efficiency of laser heating, the central wavelength of the OPO output is tuned to the local maximum of water absorption at around 1440 nm. Ultrashort (≈ 100 fs) IR pulses were used for laser irradiation^{36, 37}."

Reviewer Comments:

Reviewer #3 (Remarks to the Author):

The study is a nice piece of work and I recommend publishing it in Nature Communications.

I have a few additional comments which I added in red font in the attached file.

In the following text we omit the comments that were agreed in previous revisions and leave only the new comments of the reviewer **marked in red**. Our new responses are marked with a **blue font color**.

Comment

2. Usefulness for the neural circuit analysis community is questionable, since a lot of parameters need to be precisely controlled (see a-f below). It will still be interesting for special applications where visible light needs to be avoided, but I'd prefer if each of these potential limitations were labeled as such in the manuscript (not ordered according to importance):

a. Small dynamic range of laser powers before starting to affect non-expressing neurons with the laser (12- 40 mW, i.e. less than one order-of-magnitude, see lines 414-417).

Response

We agree with the reviewer that the dynamic range of the laser powers is quite small. However, potential limitations largely depend on the biological system used. In the experiments with *Danio rerio* the control neurons are intrinsically thermosensitive: zebrafish somatosensory neurons sense temperature along with other parameters and they express endogenous TRPV1. The key point here is that caTRPA1 enable selective laser-evoked depolarization of these neurons and we are able (yet with low dynamic range) to activate these neurons with the temperature ramps much smaller than those needed to activate neurons in the control fish. When we work with the non-heat-sensing neurons, such as mammalian cortical and hippocampal neurons, depolarization of the control neurons is more than 10-fold smaller compared to the ca- or eoTRPA1-expressing neurons responding to the same amount of the laser radiation. We have added the sentence on that to the corresponding paragraph of the Results section. Now the end of the paragraph reads as follows: "Although the dynamic range of the laser powers evoking specific vs. non-specific response is low (from 12 to 40 mW, less than 4-fold), this limitation is largely due to the intrinsic thermosensitivity of the TRPV1-expressing somatosensory neurons. This emphasizes the importance of choosing a proper channel for thermogenetic activation, which needs to open at a temperature just a few degrees above the normal animal temperature to ensure that only neurons expressing the desired channels are activated with carefully adjusted, very mild heating."

Reviewer 3: Agreed. Since we don't know for sure that the endogenous TRPV1 channels caused this effect, it would be more careful to write: "...this limitation is likely largely due to...".

Response: We have changed the text according to the reviewer's suggestion.

Reviewer 3: The duration of the stimulation phases should be indicated in the legends for Fig. 4d.

Response. Now the figure legend reads as: **(d)** Multiple cycles of neuron 1 activation by mild above-threshold heating using 20 mW laser. Pulse duration is 3 s.

Comment

3. In Figure 1, a schematic of the experimental protocol (incubation, temperature steps) should be given, so that the reader has a chance to understand the gist of the experiment by looking at the figure alone.

Response

We have added the scheme to the Fig. 1

Reviewer 3: Great.

In Fig1d, please add "30°C" to the placed text "Microscopy: heated media or AITC addition".

Response: Done

Reviewer 3: In Fig. 1b-c, please add an explanatory x-axis legend: “pre-incubation temperature”, or similar.

Response: In fact, in the figure legend we provide the detailed explanation: “36-48 hrs later cells were transferred to another incubator for temperature conditioning at 25, 30 or 37°C for 18-26 hrs. Next, half of the cells were kept for additional 1h at the same temperature and another half of the cells were kept at 25 °C. Then all cells were transferred to the stage of the microscope preheated to 25°C and activated by the addition of warm medium under control of electrode thermometer. Final temperature was 30 ± 1 °C.” We think that adding X-axis legend will overload the graph with the textual information and will duplicate the figure legend. Moreover, X-axis there does not reflect any scale. We therefore would like to keep the panels b & c of the figure as they are.

Reviewer 3: In the legend for Figure 1, the sentence “...cells were grown at 25, 30 or 37 °C and then subjected to heating from 25 °C to 30 ± 1 °C with or without 1 hour long pre-incubation at a subthreshold temperature of 25°C” doesn’t make a lot of sense, because in the case of nonincubation at 25°C two of the experimental groups are not held at 25°C and because in the case of the non-incubation-37°C-experiment, the cells are not heated but cooled to 30°C. Please rework this sentence. For example: “...cells were pre-incubated at 25, 30 or 37°C on the third day and on the fourth day cells were either kept at the same temperature as on the third day (control groups) or incubated at 25°C (experimental groups). Then all cells were incubated at 30°C under the microscope, which is above the threshold of the channel.”

In the main text, please explain the experiment better in the results section. For example, the final “heating” temperature of 30°C is not mentioned.

Response: We thank the reviewer for the attempt to make this part more clear. We have changed the legend for 1b as follows: “HEK293 cells were co-transfected with caTRPA1-IRES2-EGFP and Ca^{2+} sensor R-GECO1. 36-48 hrs later cells were transferred to another incubator for temperature conditioning at 25, 30 or 37°C for 18-26 hrs. Next, half of the cells were kept for additional 1h at the same temperature and another half of the cells were kept at 25 °C. Then all cells were transferred to the stage of the microscope preheated to 25°C and activated by the addition of warm medium under control of electrode thermometer. Final temperature was 30 ± 1 °C. Control cells express only R-GECO indicator.”

We have also made the corresponding Methods section much more clear and added temperatures to the Results section as suggested by the reviewer.

Comment

Line 254: “comparable to the size of a single cell”? This does not seem to be correct or it is unclear what you mean. The size of the cell soma is about 20 μm in the picture, three times smaller than the laser beam diameter.

Response

The full sentence cited by the reviewer is the following: " In our experiments with IR laser radiation, the diameter of the laser beam ($\approx 60 \mu\text{m}$) was comparable to the size of a single cell, thus enabling activation of individual neurons without affecting the neighboring cells (Fig. 4 a,e,f)."

The size of the cell body, $\sim 20 \mu\text{m}$, is comparable with the diameter of the beam ($\approx 60 \mu\text{m}$), not even an order of magnitude difference. As we have a deal with neurons, the effective size of the cell is larger than the size of the cell body, because axons and dendrites also have TRPA1 and contribute to stimulation of the neuron.

Reviewer 3: While the sentence is technically speaking not wrong, I still find it misleading.

Response: We have changed “comparable to the size of a single cell” to “comparable to the size of a single neuron”.

Comment

Line 275: "...the basal temperature of zebrafish maintenance in fish facilities is 26-27°C". Please provide two or three references for this claim. I could only find preferred temperatures of around 28° C in the literature. This point is quite important given the authors' claim that the fish facility temperature is sub-threshold for the channel.

Response

Temperature range used for zebrafish cultivation is wide. Various sources recommend different ranges. Our facility maintains the fish at 26-27 oC which is within optimal range (26-28.5 C) according to the Ref 48 in the text: «The room temperature or the tank temperature is generally maintained between 26-28.5 °C».

Another example:

«Temperature: The document states on p. 24, "A water temperature of 28.5° C is widely cited as the optimum temperature for breeding zebrafish. There has however, been little research to investigate the full implications of constantly keeping fish at this very specific temperature." While the document recommends a very specific temperature of 28.5° C as the optimum for breeding zebrafish, it provides a number of references with a range of 25-29° C and acknowledges that more research is needed to determine temperature preference and implication of maintaining fish at warmer temperatures for an extended period of time. Performance standards should be applied with consideration of health, welfare and speciestypical behavior." From Reed B, Jennings M (2011) Guidance on the housing and care of Zebrafish. Research Animals Department, Science Group, RSPCA 62

One more example:

«The thermal preference or optimum for zebrafish has not been formally defined. The maintenance temperature of 28.5 °C recommended by Westerfield (1995) is almost universally cited for zebrafish in culture. This optimum is supported experimentally by at least one published study, in which zebrafish showed a marked increase in growth when held at 28 ± 0 °C as opposed to lower and/or stochastically and predictably fluctuating temperatures (Schaefer and Ryan, 2006). However, based upon their wide range of tolerance in nature, it is highly probable that their actual preference range extends considerably above and below this temperature, and the wider range of 24–30 °C recommended by Matthews et al. (2002) 3 is probably more appropriate» From the Ref 50 in the text.

Moreover, similar to neurons and HEK cells, zebrafish expressing caTRPA1 can be maintained at temperatures higher than the channel threshold because of TRPA1 authoprotection mechanism an resistance to slow heating or constantly elevated temperature. They need to be cooled down to 26-27 oC before the experiment.

We have changed the sentence and provided the references:

"Since the basal temperature of zebrafish maintenance in fish facilities is generally maintained between 26-28.5oC48-50, these measurements prove that TRPA1 channels offer a powerful tool to study *Danio rerio* neurophysiology."

Reviewer 3: Yes, the new sentence is OK. But other instances in the main text should be corrected as well. Line 382 ("standard temperature") and line 513 ("normal...temperature"). As the authors also cite above, the optimum/standard temperature is rather 28,5°C and not 26-27°C (which is still an acceptable temperature). These sentences should be changed.

Response. We have changed the first sentence to the following: "Remarkably, the activation threshold of caTRPA1 in these experiments is just one or two degrees above 26 °C which is a permissive temperature for *D.rerio* maintenance, enabling *in vivo* activation of caTRPA+ zebrafish neurons using IR laser radiation."

Second sentence now reads as: : Specifically, caTRPA1 with its T_{ON} of ~28°C opens at just 1 – 2 degrees above 26-27°C that falls within the normal zebrafish maintaining temperature range."

Comment

Line 430: "muscle contraction initiation": No, the hundreds of millisecond delay are not very likely to be explainable by slow initiation of muscle contraction. E.g. during escape responses animals respond within less than 10 ms.

Response

We agree with the reviewer that this interpretation is very speculative. We removed that and now the corresponding sentence reads as "Given that the channels open, as our electrophysiological studies demonstrate, within 1 – 3 ms after the temperature reaches the channel activation threshold and that the action potential starts to build up within another 35 – 40 ms, the observed time lag of the animal response reflects the dynamics of heating."

Reviewer 3: The causality of the heating has not been shown, nor has the activation kinetics been shown for this type of cell/experiment. Therefore the claim is not strongly supported and should be toned down, e.g. "...likely reflects...".

Response. As suggested by the reviewer, we have changed this part to "...response likely reflects the dynamics of heating."

Comment

The possibility that TRPA1 activation leads to cell ablation (as shown in ref. 19 of the old ms for TRP channels) should be discussed.

Response

Well defined temperature threshold of the channel, choosing the channel with proper threshold for each biological model with the activation temperature in the physiological range, and control of the heating intensity and temperature calibration ensures that the chances of ablation are not higher than in other optical methods such as confocal microscopy or two-photon neuronal stimulation.

We have added the corresponding text in the Discussion section. Now it reads as: "Heating tissue by 1 to 3 degrees over a short period of time usually does not produce any toxic effect. Stronger heating, however, should be avoided **as potentially causing cell ablation.**"

Reviewer 3: a reference should be added with regard to TRPA1-induced ablations (e.g. ref. 19 from the old ms). It should be made clear in the last sentence that you mean TRP channel induced ablations, e.g. "Stronger heating of TRPA expressing cells, however, should...".

Response. We have changed the sentence and have added the ref 19 from the old version of the ms (now ref 21): "Stronger heating of TRPA expressing cells, however, should be avoided as potentially causing cell ablation²¹."

Reviewer 3: additional comments:

The study is a nice piece of work and I recommend publishing it in Nature Communications.

Line 52f: isn't optogenetics already a technology? Technologies □□experiments ?

Response. Agreed, changed.

Line 146: indicate the used TRPA1 construct in the main text.

Response. We have indicated the channel: "We compared the heat- and agonist-induced Ca²⁺ responses in caTRPA1-expressing HEK293 cells cultured at different temperatures..."

Line 171: why is lambda the key parameter and not the intensity? Please change this sentence.

Response. Wavelength is a key parameter because different wavelengths are differently absorbed by the sample. However, this is clearly not the only key parameter and, we agree with the reviewer, the intensity is a key parameter as well. Therefore we have changed the sentence

to the following: "The wavelength of laser radiation λ is one of the the key parameters controlling laser-induced heating of a cell and the surrounding tissue. "

Line 234: If you claim to be able to generate "desired waveforms", the reader would like to see the shape of the pre-programmed waveform (the desired one) and compare it to the one you got experimentally. But this is not how you did the experiment, I believe. So this claim should be toned down.

Response. We agree with the reviewer. The part of the sentence now reads as "various waveforms".

line 442: "snake species have evolved perfect thermosensitivity". The word "perfect" implies that the thermosensitivity has reached a final stage which cannot be improved any more. However, given our limited understanding about evolution and the biophysics of protein interaction with IR waves, it appears possible that the snake thermosensitivity could still improve in the (millions of) years to come. Therefore I would advise to avoid the word "perfect" when writing about evolutionary processes.

Response: agreed, "perfect" is removed.

Line 720ff: add a description of the control experiment.

Response. We have added the description: "For each experimental condition the control cells expressing only R-GECO1 were used in the same experimental scheme. As they did not respond to heating or AITC, all the control cells from different temperature regimens were combined into a single control group (Fig. 1b,c)."

Lines 910 to lines 913: It is not clear to me what this sentence states, please describe more carefully.

Response. We apologize for misleading text. We have tried to make the entire paragraph more clear. Now it reads as follows:

For larvae used in IR inducible movement analysis the animals were considered as response to IR if tail movement occurred not later than the 100-500 ms IR pulse ends in 10 s long record. For each IR power we produced 2-3 activations for individual embryo. Embryo was counted as responded to IR stimulation only if in all cases of activation it showed a positive response to IR stimulus. In case of negative responses (no tail movement or movement after the pulse) for a given IR power, such larvae were considered as non-responded to IR stimulation.

Legend for suppl. Fig. 9a: explain how the laser light spot was visualized (backscattered light from culture flask plastic surface?)

Response. The laser spot was visualized by the camera-imaged backscattered light, but not from the plastic surface, but rather from the surface AND the cells sitting at this surface. Pure surface produces almost no scattering and reflects at the wrong angle for our camera. We use camera that filters no IR, therefore we can image this light. We have now mentioned this in the legend of Supplementary figure 9: "Local heating of a neuron in a culture using an optical fiber. The scale bar is 100 μm . The laser spot is visualized by the camera collecting IR light backscattered from the layer of the cells."